

# Dynamic vegetation reveals unavoidable climate feedbacks and their dependence on climate mean state

Pascale Braconnot*, Nicolas Viovy and Olivier Marti

Laboratoire des Sciences du climat et de l'environnement (LSCE-IPSL, unité mixte CEA-CNRS, UVSQ, Université Paris Saclay, Orme des Merisiers, 91191 Gif sur Yvette Cedex, France

 *corresponding author : pascale.braconnot@lsce.ipsl.fr

**Abstract**. We investigate seasonal vegetation feedbacks considering mid-Holocene and pre-industrial simulations with the IPSL climate models for which dynamic vegetation is switch on. We consider four different settings for the land surface model designed to improve the representation of boreal forest. They combine different choices for bare soil evaporation, photosynthesis and associated parameters, and tree mortality. Whatever the model set up, the major seasonal differences expected between the mid Holocene and preindustrial climates remain similar, and consistent with the mid Holocene greening of the Sahara and northward shift of the northern limit of forest in the northern hemisphere. However, the way vegetation-climate interactions trigger unavoidable radiative surface albedo and water vapor feedbacks depend on the model content. Cascading feedbacks involve local snow-vegetation interactions, as well as remote water vapor and long wave radiative feedbacks in the tropics, which are needed to fulfill the global energy conservation constraint of the climate system. We show that the parameterization of bare soil evaporation is a key factor that control tree growth in mid and high latitudes. Photosynthesis parameterization appears to be critical in controlling the functioning of vegetation and vegetation-climate interactions. It affects the seasonal evolution of the vegetation and leaf area index, as well as their effect on radiative feedbacks and the sensitivity of the vegetation feedback to the climate mean state. This sensitivity needs to be considered when developing and tuning climate models.

## 1    Introduction

Green Sahara and the northern limit of forest in the northern hemisphere are key characteristics of the differences between the mid Holocene and present-day climate (i.e. Jolly et al., 1998; Prentice et al., 1996). Vegetation change during the mid-Holocene has been in the Paleoclimate Modeling Intercomparison Project (PMIP) since the beginning (Joussaume and Taylor, 1995), either to understand vegetation changes and feedback on climate (i.e. Claussen and Gayler, 1997; Texier et al., 1997) or for model evaluations purposes (i.e. Harrison et al., 1998, 2014). Interactive coupling, either asynchronously or synchronously, have highlighted some of the key feedbacks induced by vegetation changes and the way vegetation affect land albedo, soil properties or teleconnections (Joussaume et al., 1999; Levis et al., 2004; Pausata et al., 2017). These past studies have emphasized the role of the snow albedo feedback in mid and high latitudes. Typical examples concern the role of vegetation-snow albedo feedback in last glacial inception (Gallimore and Kutzbach, 1996; de Noblet et al., 1996). Fully coupled Earth System models with asynchronous coupling or online dynamical vegetation also highlight the role of indirect feedbacks of the vegetation with ocean circulation or sea-ice, amplifying the initial vegetation effect or providing a muted response in these mid and high latitudes (Gallimore et al., 2005; Otto et al., 2009b; Wohlfahrt et al., 2004). It also raises





concerns on the strength of forest-snow albedo feedbacks affecting the temperature signal in spring (Otto et al.,

39  2011).


These earlier questions are still there. In the last 10 years the increase number of transient Holocene simulations
has emphasize complementary questions on the fact that vegetation could be an important factor to consider to
reconcile the simulated temperature evolution in the early Holocene with climate reconstruction (Dallmeyer et al.,
2022; Liu et al., 2014; Marsicek et al., 2018; Thompson et al., 2022). They also raised questions on the relationship
between long term changes in vegetation, external forcing and variability (Braconnot et al., 2019), as well as on
boreal forest tipping points (Dallmeyer et al., 2021). However only a limited number of studies considers the fully
coupled climate-vegetation dynamic system in these investigations. Despite the fact that models are becoming
more complex and that there is a growing number of models with fully interactive carbon cycle, there is still a
small number of modeling groups using model configurations with fully interactive vegetation (see Arias et al.,
2021). Transient Holocene simulations with dynamical vegetation show broad agreement on the mid-Holocene
green Sahara and boreal forest, but there are still lots of discrepancies in many regional aspects of the responses
and large model biases in the representation of vegetation as the one discussed by Braconnot et al. (2019). These
difficulties come both from differences in the vegetation and land surface model (Hopcroft et al., 2017), and from
the fact that first order climate-vegetation interactions in a fully interactive system are not well understood.

Climate-vegetation feedbacks are somehow hidden in the land surface albedo and atmospheric moisture feedback
in estimates of climate sensitivity (Sherwood et al., 2020). They have a direct effect on temperature, controlling
vegetation, leaf area index, productivity, evapotranspiration, soil moisture, as well as snow and ice cover.  They
cannot be easily quantified because they depend on the mean climate state and, when it comes to simulations, to
the simulated climate mean state characteristics (Braconnot and Kageyama, 2015). A reason is that vegetation lies
at the critical zone between land and atmosphere. Its variations depend on interconnected factors such as light,
energy, water and carbon and, in turn, affect climate and environmental factors. These interconnexions makes it
difficult to disentangle the exact factors that affect the representation of vegetation in a fully interactive model.
Dynamical vegetation introduces additional degrees of freedom in climate simulations, so that a model that pro-
duces reasonable results when vegetation is prescribed might not be able to properly reproduce the full coupled
system, when climate-vegetation interactions that are neglected when vegetation is prescribed induce first order
cascading effects in coupled mode. Vegetation feedbacks are in general overlooked when developing climate mod-
els or comparing simulations performed with different models. There is thus a risk that the linkages between the
model content is not properly accounted for in model comparisons, since part of the results might come from
vegetation-climate feedbacks that can themselves be tight to the underlying mean climate state.

Here we investigate the climate-vegetation feedback in mid-Holocene and pre-industrial simulations with the IPSL
climate models using four different settings of the dynamical vegetation that combining differences in the choice
of the representation of photosynthesis, bare soil evaporation and parameters defining the vegetation competition
and distribution. The objective is to investigate the first order cascading climate-vegetation feedbacks, and to iden-
tify their dependence to the model content and the climate mean state. We first compare the mid-Holocene climate
changes obtained with the different model versions. The mid-Holocene climate is a key reference period for



paleoclimate modeling, characterized by enhanced seasonality in the northern hemisphere and reduced seasonality
in the southern hemisphere compared to present day (Kageyama et al., 2018; Otto-Bliesner et al., 2017). It is well
suited to investigate feedbacks occurring at the seasonal time scales, such as those induced by the seasonal evolu-
tion of vegetation in response to the seasonal cycle of the insolation forcing in mid and high northern latitudes. A
focus is put on the estimations of the atmospheric feedbacks resulting from surface albedo, atmospheric water
content and lapse rate following Braconnot and Kageyama (2015). We analyse the dependency of these first order
feedbacks to the representation of vegetation, and relate this to the model content, and how it affects the vegetation-
climate interactions. The aim is to highlight key factors controlling the unavoidable feedbacks induced by snow
and ice cover or by atmospheric water content through evaporation and temperature in a fully coupled system. We
also focus on the differences in these feedbacks between the model versions considering the mid-Holocene and
the preindustrial climates separately, so as to understand the dependence of the seasonal feedbacks on the climate
mean state.

The reminder of the manuscript is organized as follow: Section 2 presents the set-up of the four model configura-
tions and the suite of experiments. Section 3 is dedicated to the analyses of the differences between the mid-
Holocene and the preindustrial climates, including a quantification of the atmospheric feedbacks resulting from
the response to the mid-Holocene insolation forcing. Section 4 goes deeper in the analyses of the feedbacks con-
sidering the differences between the mid-Holocene simulations and making the linkages between these differences
and the model content, before addressing the questions of the feedback dependence to the climate mean-state. The
discussion and conclusion, section 5, highlights the key findings and the implications for the land carbon sources
and sinks.
**2    Model and experiments**
**2.1    IPSL model and different settings of the land surface component ORCHIDEE**
The reference IPSL Earth climate model version for this study is IPSLCM6-LR (Boucher et al., 2020). This model
version has been used to produce the suite of CMIP6 simulations experiments (Eyring et al., 2016), including the
mid Holocene PMIP4-CMIP6 (Braconnot et al., 2021). The atmospheric component LMDZ (Hourdin et al., 2020)
has a regular horizontal grid with 144 points regularly spaced in longitude and 142 in latitude (2.5° × 1.3°) and 79
vertical layers extending up to 80 km. The land surface component ORCHIDEE version v2.0 is run using the
atmospheric resolution. It includes 11 vertical hydrology layers and 15 plant functional types (Cheruy et al., 2020).
The ocean component NEMO (Madec, Gurvan et al., 2017) uses the eORCA 1° nominal resolution and 75 vertical
levels. The sea ice dynamics and thermodynamics NEMO-LIM3 (Rousset et al., 2015; Vancoppenolle et al., 2009)
and the ocean biogeochemistry NEMO-PICES (Aumont et al., 2015) are run at the ocean resolution. The oceanic
and atmospheric components are coupled through the OASIS3-MCT coupler (Craig et al., 2017) with a time step
of 90 mn.

Compared to the PMIP4-CMIP6 mid Holocene simulations (Braconnot et al., 2021), the dynamical vegetation
module (Krinner et al., 2005) is switch on for all the simulations considered in this study. The vegetation dynamics
is based on the approach of the LPJ model (Sitch et al., 2003). It allows to simulate the evolution of the vegetation





cover in response to climate. It accounts for several climate constraints (e.g. minimum and maximum temperature)
for vegetation fitness and competition between plant functional types (PFTs) based on their relative productivity
Starting from this reference version, two formulations of bare soil evaporation and photosynthesis have been tested.
These tests have been motivated by an underestimation of the boreal forest when using the standard version of the
dynamical module in IPCLCM6-LR (see below). The test made on bare soil evaporation uses developments de-
scribed in details in the documentation of ORCHIDEE hydrology by Ducharne et al. (2018). In the standard version
of the IPSLCM6-LR, model bare soil evaporation depends on the moisture content of the first 4 of the 11 soil
layers (Milly, 1992). The bare soil evaporation rate corresponds to the potential evaporation rate when the moisture
supply meets the demand (Cheruy et al., 2020). Another solution has been developed to better represent soil evap-
oration processes, by considering the ratio ($mc$) between the moisture in the litter zone (first four surface layers)
and the corresponding moisture at saturation (Sellers et al., 1992). With this parameterization, the aerodynamic
resistance is decreased by a factor $\frac{1}{rsoil}$ , where:
$$rsoil = e^{8.206-4.255*mc} \tag{1}$$
This adjustment in the bare soil evaporation parameterization was not incorporated into IPSLCM6A-LR due to the
fact that it induces a surface warming that was not fully understood to be used in the whole suite of CMIP6 simu-
lations (Cheruy et al., 2020).  For simplicity, the two parameterizations are respectively referred to as *bareold* and
*barenew* in the following (Table 1).

TABLE 1

The parameterization of photosynthesis/stomatal conductance used in the ORCHIDEE land surface model (Fig. 1)
is different between IPSLCM5A-LR (Dufresne et al., 2013) and IPSLCM6-LR (Boucher et al., 2020). In IP-
SLCM5A-LR (Fig 1), the photosynthesis (PhotoCM5) is represented by the standard Farquhar model for C3 (Far-
quhar et al., 1980) which has been extended to C4 plants (Collatz et al., 1992) and coupled to the Ball & Berry
stomatal conductance formulation (Ball et al., 1987). In IPSLCM6-LR (Fig. 1), the photosynthesis/conductance
(PhotoCM6) has been improved to include the approach based on Yin and Struik (2009) coupled to the original
Farqhuar (1980) model. The PhotoCM6 parameterization allows to replace the iterative resolution by an explicit
solving of the coupled photosynthesis/stomatal conductance. Important differences between the two approaches
are due to the fact that the stomatal conductance is driven by the vapor pressure deficit in PhotoCM6, whereas in
PhotoCM5 it is based on relative humidity. Also, the shape of the response of the photosynthesis to temperature is
different (Fig. 1). The temperature response is a bell shape function in PhotoCM5, which allows to control the
minimum, maximum and optimal temperature of photosynthesis independently of the maximum rate of photosyn-
thesis. The response of photosynthesis to temperature is driven by a modified Arrhenius function in PhotoCM6,
with a reference temperature of 25°C. Hence the fixed maximum rate of carboxylation $Vcmax$ it the rate at 25°C,
whereas it is the optimal $Vcmax$ in PhotoCM5 (Fig. 1), and the parameters (named ASJ) of the Arrhenius function
are prescribed. It does not allow to have a full control on the temperature response, which is the reason why we
reimplemented the PhotoCM5 parameterization to run our tests with IPSLCM6-LR. Another important difference
is that in PhotoCM6, the response to temperature is adapted to the local long term (i.e. 10 years) temperature of
each pixel whereas in PhotoCM6, the temperature dependence is fixed for the whole *pft*.





FIG1.

These differences in the shape of the function has some implication on some of the adjustments we made to the
original parameterizations to compensate for the tendency of the climate model to be too cold in some mid to high
latitude regions (Boucher et al., 2020). The objective was to allow photosynthesis at lower temperature. The pa-
rameters of photoCM6 have been adjusted using off line simulations forced by atmosphere reanalysis. The objec-
tive was to find optimal limits in temperature for PhotoCM6 and to adjust *Vcmax* at 25°C and ASJ within accepta-
ble range of values. In the standard version of the IPSLCM6-LR model these parameters are the standard ones,
and we add an *s* to the name in that case (Table 1). The photoCM5 parameterization use the standard values of
PhotoCM6.

Another important process determining the possibility for forest to grow in a cold environment is the critical tem-
perature for tree regeneration (*tcrit*). Indeed, it is assumed that, even for boreal forest, a very low temperature
during winter will induce an insufficient fitness for reproduction and then forest regeneration. In the standard
model version, it is prescribed to -45 °C for boreal evergreen needle leaf forest (pft 7) and boreal deciduous broad-
leaf forest (pft 8). It means that when daily temperature goes below 45°C a fraction of trees dies. This threshold
was too high as currently regions covered with forest regularly experience temperatures under -45 °C. We therefore
changed the critical temperature to -60 °C, the standard value used for Larix (pft 9), taking the risk to simulate a
wrong composition of boreal forest.

**2.2    Experiments**
We consider a set of four experiments (Table 1). For each of them, we performed a mid-Holocene simulation
following the PMIP4-CMIP6 protocol (Otto-Bliesner et al., 2017) as in Braconnot et al. (2021), and a pre-industrial
CMIP6 simulations (Eyring et al., 2016). The preindustrial climate we use as reference in this study has a similar
Earth's orbit configuration as today, with summer solstice occurring at the perihelion and winter solstice at the
aphelion. These experiments represent key steps in a wider range of tests designed to improve the representation
of boreal forest. Model developments were done using the mid-Holocene as a reference for natural vegetation,
knowing that the preindustrial climate is affected by land use, which is not considered in these experiments.

The different model set ups for these simulations are gathered in Table 1. The first experiment, V1, is performed
with the standard model and the dynamical vegetation switch on. The differences with the simulations presented
in Braconnot et al. (2021) are thus only due to the dynamical vegetation climate interactions. All the other exper-
iments include the new parameterization of bare soil. Experiment V2 and V3 have the PhotoCM5 parameterization
of photosynthesis. In V3 the critical temperature is modified for boreal forests. The final version V4 is parallel to
V3, using PhotoCM6 photosynthesis. Note also that some bugs and inconsistent choices when running with or
without the dynamical vegetation have been found in the standard model version after the first experiment was
completed. They have been corrected for the sensitivity tests and do not affect the results that only focus on key
factors that have emerged from a large suite of shorter systematic sensitivity experiments. Note that version V4 is
considered as the reference version for ongoing Holocene transient simulation with dynamical vegetation.






The initial state for all the simulations corresponds to a restart of the IPSLCM6-LR model for the ocean-atmos-
phere-sea-ice-icesheet system. The land-surface model starts from bare soil. We follow here the protocol used by
Braconnot et al. (2019). It guaranties the entire consistency between the simulated climate and the simulated veg-
etation. We tested that, as in our previous set of Holocene experiments with dynamical vegetation (see Braconnot
et al., 2019), the results would be the same when a vegetation map from a previous simulation is used as initial
state. This is mainly due to the fact that the land surface covers only ~30 % of the Earth and doesn't store energy
on a long-time scale, compared to the ocean. The initial state corresponds to a mid-Holocene or PI climate depend-
ing of the simulated period, except for the preindustrial simulation using V4 for which the initial state is from the
mid-Holocene simulation (Table 1).
**2.3    Vegetation-climate adjustments**
A similar sequence is found for the vegetation adjustment time in all experiments (Fig. 2). Starting from bare soil
imposes a land surface cold start, since bare soil has a larger albedo than grasses or forest. It is characterized by a
negative heat budget at the surface (Fig. 2b), a colder 2m air temperature (Fig. 2c), reduced precipitation and
atmospheric water content (Fig. 2d, e), increase sea ice volume (Fig. 2f), reduced ocean surface heat content
(Fig. 2h), large albedo (Fig. 2i) and soil moisture (Fig. 2j). There is a rapid recovery due to the fact that snow is
also absent in the initial state, so that it doesn't amplify the initial cooling. In each of the simulation the first 50
years are characterized by rapid vegetation growth, with the well-known succession of grass and trees also dis-
cussed in Braconnot et al (2019) . This first rapid phase is followed by a long-term adjustment related to slow
climate-vegetation feedback of about 300 years. As expected, the ocean heat content adjustment has the largest
adjustment time scale. The equilibrium state is characterized by multiscale variability. These interannual to mul-
tidecadal variability is smaller than the differences between the experiments, but need to be accounted for to
properly discussed differences between the simulated climatologies. A conclusion from Fig. 1 is that 300 years of
simulation is a minimum length to properly analyze the difference between the simulations, which is consistent
with the adjustment time reported by Braconnot et al. (2019). It justifies our choice to save computing time by
considering simulations from 400 to 1000 year depending on the experiment.

FIG.2
**3    Simulated changes between mid-Holocene and pre-industrial climates**
**3.1    Temperature and precipitation changes**
We first focus on the mid Holocene changes simulated by the four versions of the model, using the simulated
preindustrial climate as a reference. The major differences between the model versions are well depicted in Fig. 3
considering only annual mean surface air temperature and precipitation for the V3 and V4 model versions. During
mid-Holocene the large Earth's axial axis tilt induces a slight reduction of incoming solar radiation in the tropics
and an increase in high latitudes. This effect is further amplified (or damped) by the fact that, during mid-Holocene,
Earth's precession enhances the insolation seasonality in the northern hemisphere and decreases it in the southern
hemisphere (COHMAP-Members, 1988). The annual mean reflects thus both the annual mean change in insolation





and the large seasonal changes and the associated atmospheric, oceanic and land surface feedbacks. It is charac-
terized by an annual mean warming in mid and high latitudes in the northern hemisphere and annual mean cooling
in the southern hemisphere (Fig. 3). The annual mean cooling in the tropics over land is a fingerprint of enhanced
boreal summer monsoon (Joussaume et al., 1999). The latter is driven by dynamical effects that deplete precipita-
tion over the ocean and increase it over land (Braconnot et al., 2007; D'Agostino et al., 2019). These results are
consistent with those of the multimodel ensemble of PMIP mid-Holocene simulations (Brierley et al., 2020). They
cover a large fraction of the spread of temperature changes produced by different models worldwide (Brierley et
al., 2020), stressing that cascading feedbacks induce by dynamical vegetation have profound impact on regional
climate characteristics.
FIG. 3
The results of the different model versions are compared in Fig. 4 to those of the standard IPSL model without
dynamical vegetation and the climate reconstructions from pollen and macrofossils data by Bartlein et al. (2011).
These diagnoses complete the maps presented in Fig. 3 by indicating that the largest annual mean warming in mid
and high latitude is found for V2, and that for most of the boxes V2 simulated changes in temperature and precip-
itation are not statistically different from those simulated with V3 when accounting for uncertainties between 100-
year averages (Fig. 5). The results obtained with the version V4 are the closest to the those obtained with standard
IPSLCM6 version of the model used for PMIP4 mid-Holocene simulations. They appear to be in overall better
agreement with climate reconstructions. All versions with dynamical vegetation produce larger changes in West
Africa, as it is expected with vegetation feedback. The spread between the different 100 years differences between
the mid-Holocene and preindustrial climate for a given model version also stresses that long term variability in-
duces uncertainties in 100-year estimates of about 0.5 to 3 °C depending on the region. This needs to be accounted
for since 100-year variability can be as high as the signal in some places.
FIG. 4
**3.2    Land surface feedbacks between mid-Holocene and preindustrial climates**
These differences between the model versions come from the various feedbacks induce by the different changes
in the land-surface model and feedbacks induced by the dynamical vegetation. We synthesize the mid-Holocene
differences with preindustrial by showing the mean root mean square difference between the two climates in Fig.
5 for leaf area index (*lai*), snow, and atmospheric water content. These diagnoses allow to account both for the
differences in the annual mean and in seasonality arising in response to annual mean changes in insolation between
the mid Holocene and the preindustrial climates. In order to also account for the centennial variability, we use all
possible combinations of 100-year annual mean cycles differences between the two periods for these rms estimates,
neglecting the first 300 years of each simulation. For a given variable var in simulation 1 (*var1*) and simulation 2
(*var2*) the rms is thus computed as:
$$rms(var) = \sqrt{\frac{1}{n_1 \times n_2} \sum_{i=4}^{n_1} \sum_{j=4, j \geq i}^{n_2} \sum_{m=1}^{12} (var1 - var2)^2}$$
(2)



where $n_1$ and $n_2$ represent the number of non-overlapping 100 years in simulation 1 and 2 respectively (and ne-
glecting $n = 1$ to 3 for the first 300 years), and $m$ refers to months, with 1 being the first month of the year and 12
the last month. The dispersion between the 100-year estimates provides a measure of the uncertainty. We only
discuss in the following aspects that are statistically significant.

The *lai* rms between mid-Holocene and preindustrial climates (Fig. 5a to d) highlights that almost all regions have
change in vegetation (*lai*) at the mid Holocene compared to preindustrial. This is found with all four model ver-
sions. It also shows that regions that experience the largest changes are the Sahel Sahara, northern India, Eurasia
and the eastern part of North America, although the magnitude and regional details depend on the model version.
The large *lai* changes in Africa highlight that all of these model versions produce a green Sahara which was not
the case with the previous versions of the IPSL model (Braconnot et al., 2019). These is consistent with the in-
creased annual mean precipitation and decrease in temperature (Fig. 3). Note that this large amplification couldn't
be anticipated from the standard PMIP4-CMIP6 simulation where vegetation is prescribed to preindustrial vege-
tation, even though changes in monsoon rainfall were larger than with previous IPSL model version (Braconnot et
al., 2021). It results from vegetation feedbacks amplified by synergy with ocean feedbacks (Braconnot et al., 1999),
and from atmospheric physics and land surface improvement between the IPSLCM5 and IPSLCM6 versions of
the IPSL model (Boucher et al., 2020; Hourdin et al., 2020).

FIG. 5

Model versions producing the largest annual mean temperature changes in Eurasia and eastern north America are
also those (V3 and V4) producing the largest changes in *lai* (Fig. 5). The snow *rms* indicates that these regions
coincide with regions having the largest changes in snow cover (reduced snow cover during mid-Holocene). It is
more pronounced for the two model versions (V2 and V3) with the largest temperature changes. This is the foot-
print of a direct feedback loop between vegetation temperature and snow cover, which further triggers temperature
changes due to its large surface albedo. Mid-Holocene temperature, snow and sea-ice changes also induce sub-
stantial differences in the atmospheric water content, with largest differences arising within the tropical regions
(Fig. 5). Again, the two model versions (V2 and V3) with the largest temperature changes produce the largest
changes in atmospheric water content (Fig. 5 right (i) to (l), right column). These model versions also have the
largest changes in sea-ice between the two periods and thereby of water vapor in the north Atlantic.

**3.3    Estimations of the radiative feedbacks between mid-Holocene and preindustrial climates**
We further estimate the radiative feedbacks (Fig. 6). We quantify the shortwave (SW) radiative impact of surface
albedo, atmospheric diffusion and scattering on the Earth radiative budget at the top of the atmosphere using the
simplified method developed by Taylor et al. (2007). It consists in estimating the integral properties of the atmos-
phere (scattering, diffusion) and the footprint of the surface albedo on the top of the atmosphere shortwave radia-
tions for the different climates. Following Braconnot et al. (2021), we first estimate for each simulation the atmos-
pheric absorption $\mu$ as:
$$\mu = \alpha_p \left( \frac{SW_{si}}{SW_i} \right) (1 - \alpha_p) \tag{3},$$



and the atmospheric scattering $\gamma$ as:
$$\gamma = \frac{\mu - \left(SWsi / SWi\right)}{\mu - \alpha_s\left(SWsi / SWi\right)} \qquad (4),$$
where $\alpha_p$ and $\alpha_s$ stand respectively for the planetary and the surface albedos, and $SW_i$ and $SW_{si}$ for the incoming
solar radiation at the top of the atmosphere (insolation) and at the surface. The planetary and surface albedos are
computed from the downward and upward SW radiations. By replacing one by one the factors obtained for one
climate (or one simulation) by those obtained for the other climate (or another simulation) we have access to the
radiative effect of this factor between the two climates (or two simulations). As an example, the effect of a change
in the surface albedo in simulation 2 compared to simulation 1 used as reference is provided by:
$$\alpha_p\left(\mu_1, \gamma_1, \alpha_{s_2}\right) - \alpha_p\left(\mu_1, \gamma_1, \alpha_{s_1}\right) \qquad (5)$$
The decomposition done for short wave radiation is not valid for long wave (LW) radiation (Taylor et al., 2007).
However, in the case of the simulations considered here we can assume that the LW forcing due to trace gazes is
small (Braconnot et al., 2012; Otto-Bliesner et al., 2017). The mid Holocene change in outgoing longwave radia-
tion at the top of the atmosphere (TOA) corresponds thus to the total LW radiative feedbacks. The outgoing long
wave at TOA is composed of two terms, the surface outgoing longwave radiation ($LWsup \sim \sigma T^4$, where $\sigma$ is the
Stefan-Bolzmann constant) associated to the surface and the atmospheric atmospheric heat gain ($LW_{sup} - LW_{TOA}$)
resulting from the combination of changes in atmospheric water vapor, clouds and lapse rate. The relative magni-
tude of these different terms cannot be estimated here.

FIG. 6

We focus for this feedback quantification on the mid to high latitudes between 45° N and 80° N where differences
in *lai* and in snow cover are the largest between the mid-Holocene and the preindustrial climates between the
simulations (Fig. 6). The V2 and V3 versions of the model produce feedbacks as large as the forcing, except it is
maximum in boreal spring when the forcing is maximum in summer and early autumn. The dominant factor to
amplify the insolation forcing is the land surface albedo (Fig. 6b). It results from the combination of vegetation
and snow changes, with a dominant effect of snow because of its larger albedo. The snow albedo effect is amplified
when grass is replaced by forest in the mid-Holocene simulation, which occurs over a large area in Eurasia for V2
and V3 compared to V1 where grass is dominant or V4 where a larger fraction of forest is still present in the
preindustrial simulation (Fig. 7). Feedbacks in LW radiation have also a large impact in modifying the top of the
atmosphere total radiative fluxes. It reduces the effect of the albedo feedback by allowing more heat to escape to
space. Interestingly, the direct surface temperature effect (Planck) is partly compensated by an increased green-
house gas effect resulting from increased water vapor and change in atmospheric lapse rate, in places where the
surface warming is maximum (Fig. 3 and Fig. 5).

FIG. 7
**4    Differences between model versions and dependence of radiative feedback to climate mean state**
The first order feedbacks highlighted between vegetation, temperature, snow and albedo in previous section have
different magnitude depending on the model versions (Fig. 6). They arise from differences in model content and





first-order albedo and water vapor feedbacks, some of which may mask the initial effect due to model content. We
thus investigate if we can attribute some of the systematic differences in climate and vegetation cover to the dif-
ferent parameterizations and tuning of bares soil evaporation, photosynthesis or pft 7 and 8 critical temperature.

### 4.1 Systematic differences between model versions for the mid-Holocene

The successive model developments were targeted to produce mid-Holocene boreal forest as the dominant pfts
further north in Eurasia and north America when going from the V1 to the V4 versions of the model (Fig. 7).
Considering only the dominant pft in Fig. 7 masks the fact that vegetation is represented by a mosaic of 15 pfts in
each model grid box. We present in Fig. 8 the global vegetation assemblages of the 15 pfts in order to better
highlight the differences between the simulations. It reflects the major differences found at regional scales. As
expected from model developments, major differences between the simulations are found for pft 7 (Boreal
Needleleaf Evergreen) for the mid-Holocene (Fig. 8a). It represents about 5-10% of the total vegetation cover in
V1 and V2 and 13% in V2 and V4. In V2, pft 9 (Boreal Needleleaf Deciduous) is the dominant type of boreal
forest (9% of total vegetation cover), while in V1, boreal forest is poorly represented.

FIG. 8

All model versions, except V1, use *barenew* parameterization for bare soil evaporation. It appears to be a critical
model aspect contributing to a better representation of boreal forest. Bare soil evaporation is small in all simulation
except V1 where it peaks in May-June (Fig. 9a), at a time when tree leaves are growing in the northern hemisphere
and soils are saturated. With the *bareold* parameterization the evaporation in these conditions is close to potential
evaporation. The other simulations do not produce the large boreal spring bare soil evaporation, due to the fact
that evaporation is limited by soil and biomass characteristics (see Section 2.1).  In these simulations, the evapo-
transpiration is slightly larger and peaks in July-August at the time of the maximum development of vegetation in
the northern hemisphere (Fig. 9a). Statistically significant higher values are found for V4 which is also the warmest
simulation. As a result, surface soil moisture is larger in V2 to V4 compared to V1, and favors tree growth. Inter-
estingly the total evaporation remains almost the same between all simulations but the surface soil moisture is
higher in V3 to V4 compared to V1 (Fig. 9c).

FIG. 9

Large differences are also found in the distribution between the different grass pfts between the mid Holocene
simulations, V1 having the largest proportion of pft 10 and 14, which results and contribute to the fact that this
simulation is the coldest one with the largest snow cover (Fig. 9b). The partitioning between grass and tree leads
to different magnitude in soil moisture between the simulations, associated to difference in root depths and to the
way these different types of vegetation recycle water. It affects temperature through evaporative cooling and the
amplification of the surface albedo by snow in mid and high latitude. The differences in the surface albedo and the
different linkages between snow and vegetation are well depicted in Fig. 10 (a) for the 45°N-80°N region. The
large difference found between the simulations for albedos in the range 0.3 to 0.7 is the footprint of the difference
in the ratio of tree and grass cover, with grass dominant vegetation for V1 and a mixture of grass and pft 9 for V2.





The peak emerging for albedo around 0.22 in V3 and V4 is related to the pft 7 coverage in these simulations
(Fig. 8). It highlights that the overall albedo combination of tree and snow albedo leads to a smaller albedo in these
two simulations. All the mid-Holocene simulations have a quite similar coverage of high albedo, which is com-
patible with similar distribution of sea-ice and regions fully covered by snow. The lower coverage for albedo >
0.7 % is for V4 which has the smaller sea-ice cover.

FIG. 10

In terms of radiative feedbacks between 45° N and 80° N, the surface albedo effect varies significantly between
the simulations (Fig. 12a, b). The radiative feedbacks are computed using the Taylor et al. (2009) methodology
following what was done for the mid-Holocene differences with the preindustrial climate in section 3.2, except
that the V4 version of the model serves as reference. Positive values (negative) indicate that the feedback brings
more (less) energy to the climate system in V4. Since we compare the simulations for a given climate, the forcing
is the same for all the mid-Holocene simulations, and the only factor affecting the global energy balance comes
from differences in seasonal climate feedbacks. The largest differences in surface albedo feedbacks between the
V4 and the V1 to V3 versions of the model occur from February to July. It is maximum for V1 due to the largest
snow-vegetation albedo feedback, as expected from the distribution of surface albedo between 45° N and 80° N
(Fig. 10). This effect exceeds 10 W m$^{-2}$ (up to about 16 W m$^{-2}$) from April to June. The effect is smaller between
V4 and V2 or V3, but with maximum of 10 W m$^{-2}$ for V2 and 8 W m$^{-2}$ for V3. Note that for V2 and V3 the larger
radiative effect comes from the albedo combination with the type of boreal forest (pft 7 or pft 9) more than from
the relative distribution between grass and forest (Fig. 10).

The cloud SW feedback differences between the simulations slightly amplify the effect of the surface albedo from
April to September in V1 to V3 compared to V4 (Fig. 11c). Part of the signal is damped by long wave radiation
resulting from temperature, clouds and lapse rate (Fig. 11d). This is mainly due to the differences in evapotran-
spiration between the mid-Holocene simulations (Fig. 9a) resulting from the combination of vegetation character-
istics, but also from differences from the insulating effect of snow and ice cover in mid and high latitudes (Fig. 11).
It is strongly tied to temperature, and thereby to the atmospheric water holding capacity. The V4 mid-Holocene
simulation has the largest atmospheric water content, with maximum difference with the other simulation in the
northern hemisphere (Fig. 12). Despite the fact that the representation of boreal forests and the interactions between
vegetation and snow (Fig. 8, and 10) is the major cause of the differences between the simulations, the largest
water content differences are found in the tropics, with statistically significant differences found up to 40° S (Fig.
12). It reminds us that, in the fully coupled system, rapid energy adjustment between the hemispheres and between
land and ocean are induced by the regional differences in energy sinks and sources, and that these rapid telecon-
nexions also shape the simulated climate mean state. It also stresses the important role of the tropics and tropical
ocean in regulating the global atmospheric moisture, and in balancing solar forcing and SW feedbacks.

FIG. 11





The role of photosynthesis in regulating seasonal feedbacks needs to be highlighted. An example of systematic
difference in vegetation cover associated to PhotoCM5 and PhotoCM6 is found in Australia where the dominant
pft is forest for V3 and V4, whereas it is grass for V1 and V2. It doesn't affect much the representation of boreal
forest which is quite similar between V3 and V4, certainly because the parameter adjustments were done with the
aim to allow tree growth for lower temperature than the one used in the standard model version (see section 2.1).
At the global scale, despite different distribution of vegetation, the two simulations with PhotoCM5 (V2 and V3)
instead of PhotoCM6 (V1 and V4) exhibit a larger *lai* seasonal cycle (Fig. 9e), whatever the realism of the simu-
lated vegetation. In V2 and V3, GPP has a strong increase from March to July when the peak GPP is reached (Fig.
9f). The *lai* seasonality is smoother in V1 and V4. The parallel *lai* seasonal evolution between V1 and V4 reflects
a similar behaviour with an offset resulting from differences in temperature and differences in vegetation coverage
(in particular for temperate forests). The shape of PhotoCM5 as a function of temperature compared to PhotoCM6
(Fig. 1) favours larger productivity (*gpp*) as soon as *lai* is developing. This means that for given climatic condi-
tions, the start of the growing season should be similar with the two parameterisations, but photoCM5 should have
larger *gpp*. This is indeed what we obtained between the simulations (Fig. 9e, f). This systematic difference affects
the seasonality of the surface albedo, through the *lai* and the total soil moisture. Reduced *gpp* during the growing
season in the northern hemisphere implies more humidity in the soil, as it can be seen on Fig. 9 between V4 and
V2 or V3, which are simulations sharing the same bare soil evaporation. Due to all the interactions in the climate
system, we also end up with the counter intuitive result that V4 has the largest vegetation cover (Fig. 9), but that
the vegetation is less productive than in V3 and even V2.

FIG. 12
**4.2**     **Dependence of vegetation induced radiative feedbacks on mean climate state.**
The feedbacks and their seasonal evolution between the model versions discussed for the mid-Holocene are very
similar to those occurring between Mid-Holocene and preindustrial climates for each model version. However, the
comparison of Fig. 6 and Fig. 11 (a to d) suggest that the strength of the feedback is different between the two
periods, which is indeed the case (Fig. 11e to h). It raises questions on the way to compare different periods and
use them to investigate non-linear effects and thresholds.

The simulations have all in common consistent changes in vegetation between the mid-Holocene and preindustrial
climates (Fig. 7 and 8). At the global scale the larger fraction of bare soil and grasses simulated for the preindustrial
climate (Fig. 8) is consistent with the drying of the Sahara Sahel, and the southward retreat of the tree line in the
northern hemisphere (Fig. 7). Also, most of the inter model differences in vegetation cover discussed for the mid-
Holocene are also found between the simulations of preindustrial climate (Fig. 7 and 8). In particular, this is the
case for the representation of the mosaic vegetation at the global scale (Fig. 8b), so that magnitude of change
between the two periods for each pft is consistent with the distribution of vegetation for each model version (Fig.
7 and 8). The distribution of vegetation appears thus to first order as a factor characterizing a model version.

However, notable differences are found that have implications on the relative differences between the preindustrial
simulations when considering the preindustrial quantification of the 45° N and 80° N SW and LW radiative




feedbacks (Fig. 11). It highlights that the sensitivity of the feedback to the climate mean state is higher for model
versions with PhotoCM5 rather than PhotoCM6 (Fig. 11). The differences reach up to 20 W m$^{-2}$ in V4 compared
to V2 to 25 W m$^{-2}$ in V4 compared to V3. It is in part due to a larger impact of snow albedo (Fig. 10). For example,
V3 and V4 have a similar fraction of pft 7 in the mid-Holocene, but not in the pre-industrial climate (Fig. 8). In
V3, boreal forest is replaced by a larger fraction of grass (pft 11) and bare soil (pft 1). There is also a larger fraction
of grass and bare soil in V2, whereas V1 doesn't change much compared to the mid-Holocene and vegetation is
dominated by grass and bare soil (Fig. 8). There is thus a larger fraction of the points where the surface albedo is
in the 0.3 to 0.7 range (Fig. 10).  Contrary to the mid-Holocene climate, there are also large differences in the 0.7
to 0.9 albedo range characterizing snow and sea-ice between the simulations (Fig. 10b). The largest increase is
found for V2 and V3, for which the number of points with low albedo value is also reduces, confirming that it is
due to a larger increase in sea-ice cover in these two simulations. For the preindustrial period, all simulation, except
V4 have a two large cover of sea-ice over the ocean (not shown) and cold temperatures associated to it. The initial
vegetation-albedo feedback is amplified by the sea-ice albedo feedback, which affect temperature, water vapor,
and the crossing of different thresholds controlling vegetation growth.

The difference in the seasonal insolation forcing compared to the mid-Holocene induces differences in the shape
of the surface albedo feedback in model differences as a function of month, with values that are still large between
V4 and V2 or V3 from July to September (Fig. 11b, f). Differences between the two periods are also found in the
seasonality of atmospheric scattering (mainly due to clouds), even though the magnitude of the scattering is quite
similar to what was obtained for the mid-Holocene (Fig. 11c, g). As for the mid-Holocene, the seasonal insolation
and temperature at the beginning of the growing season trigger some of the important differences in the photosyn-
thesis and gpp depending on the photosynthesis parameterization. The major differences in the relationship be-
tween *lai* and gpp discussed for the mid-Holocene (Fig. 9e, f) are also found for the preindustrial climate (not
shown). However, *lai* is more similar between the V3 and V4 because of larger grass and bare soil fraction in V3,
which compensate from the tendency to produce larger *lai*. The difference in vegetation growth between V3 and
V4 is controlled by the critical threshold for tree mortality and the shape of the photosynthesis curve. Compared
to the mid-Holocene climate, less insolation is received in mid and high latitude during boreal summer. For both
climates the simulation using PhotoCM5 is colder. Therefore, the surface temperature is closer to the *tcrit* value
in Spring in this simulation, compare to the simulation using PhotoCM6. It induces a larger reduction of the tree
cover and of *lai* and *gpp* (not shown), In contrast, the pre-industrial vegetation growth follows the seasonal inso-
lation forcing as for the mid-Holocene climate with PhotoCM6. This result implies that the sensitivity of the sea-
sonal vegetation feedback is a critical factor that needs to be properly constraint to reduce uncertainties in radiative
feedbacks and vegetation-climate interactions and associated cascading effects.

The cascading effects involve the LW radiative feedback and its linkages with temperature (Fig. 11h). The larger
longwave radiative feedback in V4 is accompanied by a larger atmospheric heat gain to compensate for the larger
shortwave radiative feedback compared to the mid-Holocene (Fig. 10).  As for the mid Holocene, the atmospheric
water vapor heat content is larger for V4 compared to the other simulations with larger differences found in the
tropics and in the northern hemisphere. The amplification in the atmospheric water content in northern hemisphere
reflect the differences in sea-ice cover and thereby evaporation over the ocean. Interestingly the difference in



atmospheric water content is similar in the preindustrial and mid-Holocene simulations between V4 and V1
whereas it is a factor 2 in the preindustrial compared to the mid-Holocene between V4 and V2 or V3. It is clearly
tied to the amplitude of vegetation changes and sea-ice feedback. The large differences in annual mean temperature
in mid and high latitude between the mid-Holocene and preindustrial climates simulated with the different version
of the model (Fig. 3) come thus for a large part from the simulations of the preindustrial climate.

## 5    Discussion and conclusion

The suite of mid-Holocene and preindustrial climate simulations considered here allow us to dig into the complex-
ity of the Earth's climate system. We insist on the fact that climate-vegetation interactions induce seasonal feed-
backs that trigger unavoidable first order albedo and water vapor radiative feedbacks. A full understanding and
thereby the ability to improve Earth's system model simulations requires studying these feedbacks in the fully
coupled system. Indeed, the climate mean characteristics is related to the relative contributions of SW and LW
radiative feedbacks that are needed to balance the top of the atmosphere radiative budget depending on the forcing.
These feedbacks do not necessarily occur where changes occur on the land surface, but remotely, as it is the case
for the water vapor in this study, which is maximum in the tropical regions when major snow-ice-vegetation albedo
feedbacks are maximum in mid and high northern latitudes (Fig. 5 and Fig. 11). The LW radiative feedback is less
discussed when the role of vegetation is inferred from vegetation alone simulations or simulations where the sea
surface temperature and sea-ice cover are prescribed. It is a first order effect associate to the change in temperature
and fulfil the convective radiative equilibrium which serves as a basis for the reasoning on climate sensitivity
(Dufresne and Bony, 2008; Manabe and Wetherald, 1975; Sherwood et al., 2020). It is often neglected also because
it is maximum over the ocean and in the tropics, which is in general not part of the focus when analysing vegetation
over land.
FIG. 13
We show that dynamical vegetation reveals how the land surface and seasonal evolution of vegetation trigger
atmospheric feedbacks, considering the mid-Holocene and the preindustrial climates. The comparison of two pe-
riods that have no difference in annual global mean forcing, but difference in seasonality induced by Earth's orbit
provides an efficient way to investigate the role of the vegetation seasonal feedbacks and how they are affected by
bare soil evaporation, photosynthesis and temperature threshold for boreal tree mortality. It also allows to investi-
gate the sensitivity of vegetation feedbacks to the mean state dependence. We synthetize the results considering
global annual mean in Fig. 13.  At the global scale the warmest mid-Holocene simulation is the one run with V4.
There is almost 1 °C difference with the coldest simulations, V1. As expected from the Clausius Clapeyron rela-
tionship, since the atmospheric and oceanic physics are the same between the model versions, the warmest simu-
lation is also the simulation with the highest atmospheric water content. The warmest simulation has also the
smallest snow and sea-ice cover (Fig 13d, e). The step changes between the model version for snow cover and sea-
ice cover is different from the one in temperature or atmospheric water content, which we attribute to the fact that
the seasonal vegetation-albedo feedback depends on the mosaic vegetation and its associated *lai* and productivity
(Fig. 13f to k).



Model content and vegetation-climate temperature interactions lead to different vegetation cover, with maximum
difference in the representation of gasses, temperate and boreal forest for the mid-Holocene simulations (Fig. 13).
Our results show that the warmest simulation is not necessarily the one with the largest *lai* and productivity. The
latter are mainly driven by the choice of the photosynthesis parameterisation (section 4). The climate state depend-
ence of the seasonal feedbacks is also driven by the differences in the photosynthesis parameterisation. The pa-
rameterisation, and the way it triggers vegetation growth and gpp, regulate the strength of the snow-vegetation
feedbacks and its functioning when temperature reaches the threshold temperature for tree mortality. This is inde-
pendent of the exact representation of the vegetation cover.  Differences in the magnitude of the seasonal feedbacks
leads to different directions in the global annual mean temperature values between mid-Holocene and preindustrial
periods. Vegetation cover is reduced in the preindustrial climate compared to the mid-Holocene, which is a well-
known fact (Bigelow et al., 2003; Jolly et al., 1998; Prentice et al., 1996). However, depending on the choice of
the photosynthesis parameterisation *lai* and *gpp* are reduced or slightly increased in the preindustrial (Fig. 13k, l),
which characterises a larger seasonal sensitivity for photoCM5 than photoCM6. Vegetation feedbacks are such
that the global mean temperature is reduced with photoCM5, and the increase of snow and sea-ice cover is larger.
Temperature is slightly increased, and global snow cover similar to mid-Holocene with photoCM6 and sea-ice
slightly increased. The sea-ice and snow feedbacks are first triggered by the seasonal insolation forcing, but are
amplified by land-surface climate interactions. The bare soil evaporation doesn't affect the direction of the annual
mean changes between mid-Holocene and preindustrial climates. It has a major impact on the tree cover, and
thereby on temperature though snow-vegetation-evaporation feedback.

Our results confirm that the simulated vegetation is an integrator of the seasonal feedbacks and is fully representa-
tive of the climate annual mean state. This result is somehow trivial since it is in full agreement with the definition
of climate in geography that involve the linkage between weather and environment. It is valid between climates,
but also between simulations run with different model versions (Fig. 7 and Fig. 8). Dynamical vegetation is thus a
key factor to consider to infer the realism of vegetation feedback in the climate system. These feedbacks cannot
be fully inferred from simulations where vegetation is prescribed. In addition, our results point to important dif-
ferences induced by photosynthesis that can only be assessed in the fully coupled system. They also indicate that
seasonal feedbacks have a key impact on climate changes. Paleoclimate periods for which the major difference
with present day come from the annual cycle of the insolation forcing such as the mid-Holocene or the last inter-
glacial periods considered as part of PMIP (Kageyama et al., 2018; Otto-Bliesner et al., 2017) are well-suited to
provide observational constraint on these feedbacks, even when indirectly from seasonal information on tempera-
ture, precipitation, sea-ice cover, or from vegetation.

The global annual differences between the simulations are small, even though statistically significant, and comes
from differences in the simulated climate annual mean cycle. It stresses that a proper evaluation of climate varia-
bles cannot be properly infer from annual mean values, and that specific time in the year when key feedback occur
need to be targeted in order to go one step further. This is certainly also true for climate reconstructions. Depending
on the method and records considered, substantial differences are found in annual mean reconstruction for the mid-
Holocene climate (Brierley et al. 2020). The choice of records and physical or biogeochemical variable should
thus be chosen depending on the feedback or process considered. Our results also highlight that considering multi-





decadal to centennial variability is needed because it can be high in some regions or simulations. This has been
discussed for a long time for paleoclimate simulations (Hewitt and Mitchell, 1996; Otto et al., 2009a), but model
groups tend still to only provide simulations with limited length when contributing to the PMIP database for model-
intercomparison (Brierley et al., 2020). This might lead to erroneous model ranking or interpretation of model
differences in some cases.

The reference period chosen to evaluate the results of the simulated vegetation is also an issue. In this study, we
show that differences between the model version in the simulation of mid-Holocene climate and vegetations
changes come mainly from the simulation of the preindustrial climate. Indeed, differences are larger between the
simulations of the pre-industrial climate than between the simulations of the mid-Holocene climate. Direct com-
parison of the mid-Holocene climate, and not the differences with pre-industrial, would be required, as well as
differences between different climate periods, to fully infer the realism of the simulated climate. This is a direction
to consider for future research that would help better infer the ability of a model to simulate the annual mean cycle.
The reference period is also an issue to evaluate the simulated vegetation. We directly develop the model version
using simulations of the mid-Holocene climate. The V3 and V4 version of the model appears to be rather equiva-
lent with respect to the simulated vegetation, except that a full evaluation was not done. This would requires
transforming the 15 simulated pft into the equivalent biomes inferred from pollen (Prentice et al., 1996) which was
out of the scope of this paper and also introduce artificial choices (Braconnot et al., 2019; Dallmeyer et al., 2019).
It is also difficult to compare the preindustrial simulations with preindustrial vegetation maps, because land used
is not considered in these simulations. Land use has also an indirect impact on the simulated natural vegetation
through its effect on temperature and evapotranspiration. An attempt to evaluate the simulated vegetation is pro-
vided on Fig. 9 by considering only grid points for which there is no land use in the preindustrial climate. The stars
on the figure correspond to the fraction occupied by each pft for the V4 version and the reference 1850 vegetation
map used when vegetation is prescribed to the model, as it is the case for CMIP6 preindustrial simulations (Bou-
cher et al., 2020). It suggests that, for the preindustrial climate, this last version of the model overestimates the
fraction of tropical forest (mainly pft 3), has a reasonable representation of temperate forest (pft 4 to 6), overesti-
mates boreal forest (mainly pft 7), and has a reasonable representation of grass, with the caveat that there is a
misbalance between pft 15 and pft 11.  Overall it is quite reasonable and better than in the other versions (not
shown). This model version has been retained for transient Holocene simulations with the IPSL model and dy-
namic vegetation.

FIG. 14

Our results have also implications for the land-surface carbon feedbacks and the representations of the interactions
between energy, water and carbon cycle in Earth system models. Here the carbon dioxide concentration is pre-
scribed in the atmosphere, but the carbon cycle is activated, so that carbon fluxes between the surface and the
atmosphere can be diagnosed. The pattern and magnitude are model version dependent. Figure 14 illustrates the
differences that are found between the mid-Holocene and the preindustrial climates, considering the V3 and V4
version of the model. As expected, the differences induced by the photosynthesis on *gpp* and climate lead to sig-
nificant differences in the mid-Holocene change in carbon fluxes over land. It would lead to differences in regional



and global carbon concentration in the atmosphere if carbon was fully interactive, and thereby certainly to different
climate and vegetation characteristics. This need further investigation. It also stresses that further emphasize on
seasonality is needed to better assess land-surface feedbacks and the way they trigger the first order albedo and
water vapor radiative feedbacks. Clouds certainly need to be also considered. They are key component and a major
source of uncertainty when considering different climate models with different atmospheric physics, but are of an
order of magnitude smaller that the other two feedbacks and their uncertainties when considering only the linkages
with seasonal vegetation feedbacks, as it is the case here.
Finally, a further implication of this study it that dynamical vegetation is an important factor in the climate system
and should be considered in Earth System Model (i.e. climate models with interactive carbon cycle). simulations
used to investigate possible futures if one which so properly account for the way land surface triggers cascading
feedback effects in a changing climate. This also means more degrees of freedom in the system, and thereby to
potentially larger model biases or uncertainties despite more accurate representation of internal processes.
**Data availability:** All data used to produce the different figures have been posted on the FAIR repository under
https://doi.org/10.5281/zenodo.14536307.
**Author contribution**: All authors contributed to the experimental design. PB and NV developed and implemented
the necessary changes to the land surface model. PB developed and run the coupled simulations. PB and OM
performed the analyses of the coupled simulations. All authors contributed to the drafting of the manuscript.
**Competing interests**: The authors declare that they have no conflict of interest.
**Acknowledgements**. It benefits from the development of the common modeling IPSL infrastructure coordi-
nated by the IPSL climate modeling center (https://cmc.ipsl.fr). Data files were prepared with NCO (NetCDF
Operators; Zender, 2008, and http://nco.sourceforge.net). Maps were drawn with pyFerret, a product of
NOAA's Pacific Marine Environmental Laboratory (http://ferret.pmel.noaa.gov/Ferret,). Other plots are
produced with PyFerret or with Matplotlib (Hunter, 2007, and https://matplotlib.org) in Jupyter Python note-
books.
**Financial support**: We acknowledge the project TipESM "Exploring Tipping Points and Their Impacts Using
Earth System Models". TipESM is funded by the European Union. Grant Agreement number: 101137673.
DOI: 10.3030/101137673. Contribution nr. 6. This work was granted access to the HPC resources of TGCC
under the allocations A0170112006, A0150112006, A0130112006, and A0110112006 made by GENCI.

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



**6 Table**

| Configuration name | Land surface model configuration | Period | Initial state Ocean + atmosphere | Length |
|---|---|---|---|---|
| **V1** (Vdyn00) | bareold, photoCM6s | H6k | As for the IPSLCM6 PMIP4 simulation* | 1000 |
| | | PI | Year 1870 of IPSLCM6 PI simulation [+] | 1000 |
| **V2** (Vdyn17) | barenew, photoCM5 | H6k | As for the IPSLCM6 PMIP4 simulation* | 600 |
| | | PI | Year 1870 of IPSLCM6 PI simulation [+] | 400 |
| **V3** (Vdyn21) | barenew, photoCM5, *tcrit* | H6k | As for the IPSLCM6 PMIP4 simulation* | 600 |
| | | PI | Year 1870 of IPSLCM6 PI simulation [+] | 700 |
| **V4** (Vdyn28) | barenew, photoCM6, *tcrit* | H6k | As for the IPSLCM6 PMIP4 simulation* | 1000 years |
| | | PI | Year 300 of V4 H6ka | 500 year |


Table 1. Characteristics of the different simulations. The different columns refer to the name of the simulation,
considering the name in this paper V1 to V4 and our internal simulation number (in parentheses), the initial state
and length of the simulation of the mid-Holocene (H6ka) and the preindustrial (PI) simulations. Only the initial
state for the ocean-ice-atmosphere component is provide, since all simulations, except V4 PI, start from bare soil.
*see Braconnot et al (2021), [+] see Boucher et al. (2020).






**Figures**


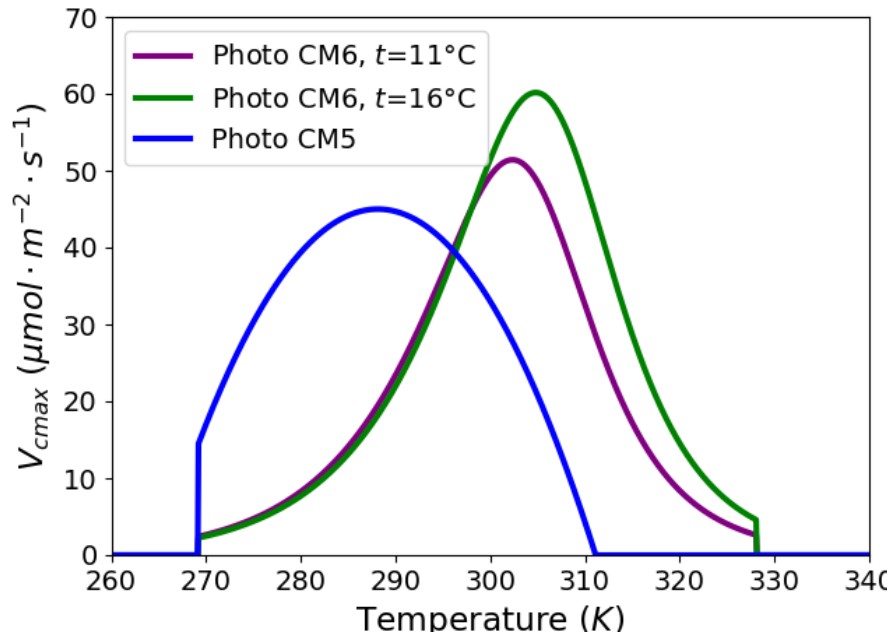


Figure 1: Maximum rate of carboxylation (vcmax, mmol m$^{-2}$ s$^{-2}$) as a function of surface air temperature (K) for
the two photosynthesis parameterization (photoCM5 and photoCM6) and *pft7*.Since phtosynthsis has a depend-
ence on the long term mean monthly temperature, the vcmax curves are plotted toe a mean temperature of 11 and
16°C. Note that with the choice we made, vcmax at 25°C for photoCM6 and the maximum value of photoCM5
are the same. See text for details on the parameterizations.


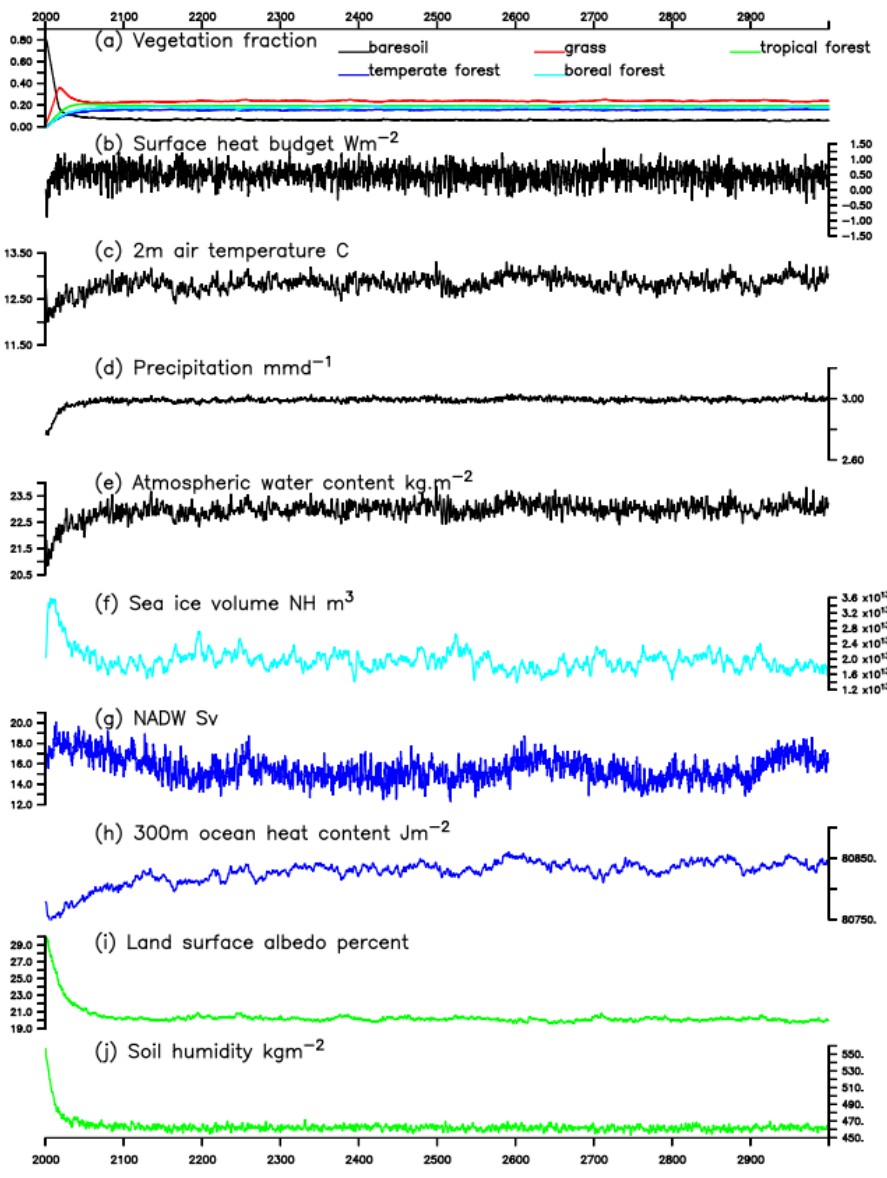


Figure 2. Adjustment time for the V4 mid-Holocene simulation, considering (a) the coverage (fraction) of 4 major

vegetation types (grass, tropical forest, temperate forest, and boreal forest) and bare soil, and a subset of climate

variables in the atmosphere (black), including (b). surface heat budget (W m⁻²), (c) 2m air temperature (°C), (d)

precipitation (mm d⁻¹), (e) atmospheric water content (kg m⁻²), in the sea ice (light blue), including (f)in sea ice

volume (m³) in the northern hemisphere (NH), in the ocean (blue), including (g) the North Atlantic Deap Water

formation (NADW, Sv) and (h) the surface 300 m heat content (J m⁻²), and in the land surface (green), including

(i) the land surface albedo (%) and (j) the soil humidity (kg m⁻³) (see text for details on the initial state).




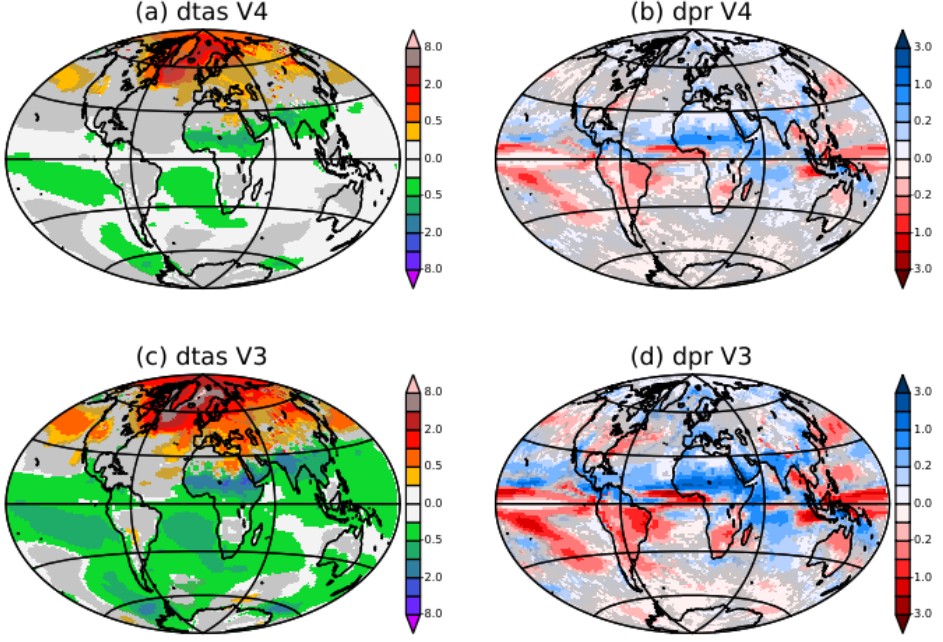

Figure 3. Simulated mid-Holocene (MH) minus Preindustrial (PI) differences for (a) and (c) the 2m air temperature
(°C) and (b) and (d) the precipitation (mm d$^{-1}$) and (a) and (b) the V3 and (c) and (d) V4 model versions. Changes
are considered to significant at the 5% level outside the grey zones. The significance is estimated from all combi-
nations of differences between 100-year averages between MH and PI simulations. For these estimates the first
300 years of the simulations are excluded.





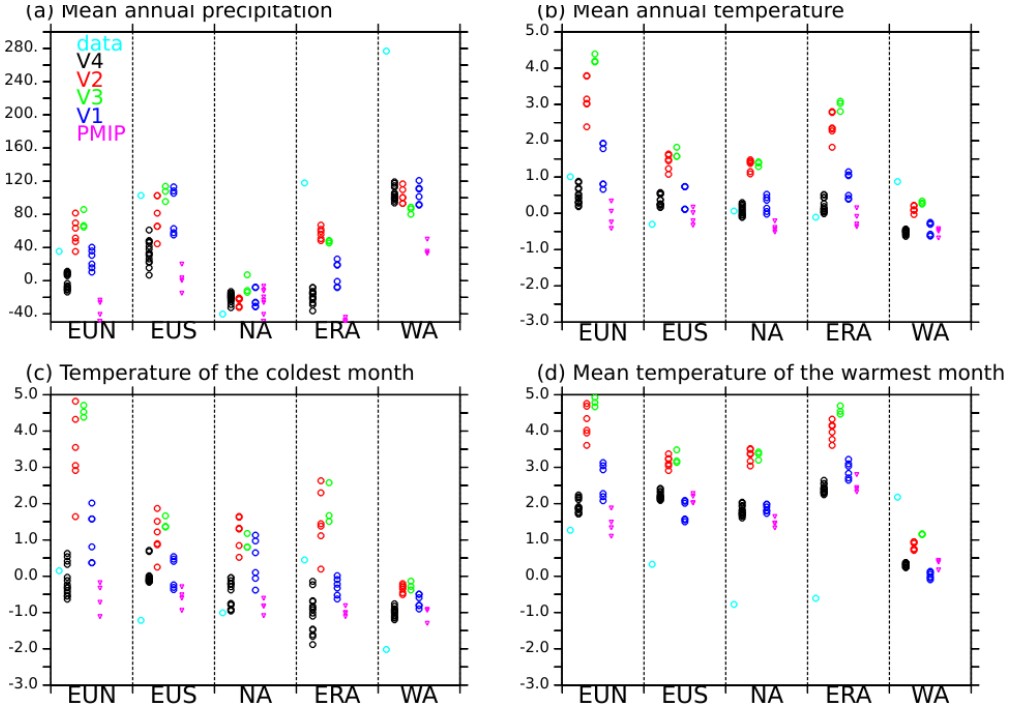

Figure 4. Comparison of the simulated MH minus PI differences with Bartlein et al. (2011) reconstructions for (a) the annual mean precipitation (mm yr⁻¹), (b) the annual mean temperature (°C), (c) the temperature of (c) the coldest month(°C) and (d) the warmest month temperature (C) and 5 selected regions for which the data coverage is high: Northern Europe (EUN), Southern Europe (EUS), North America (NA), Eurasia (ERA) and West Africa (WA).



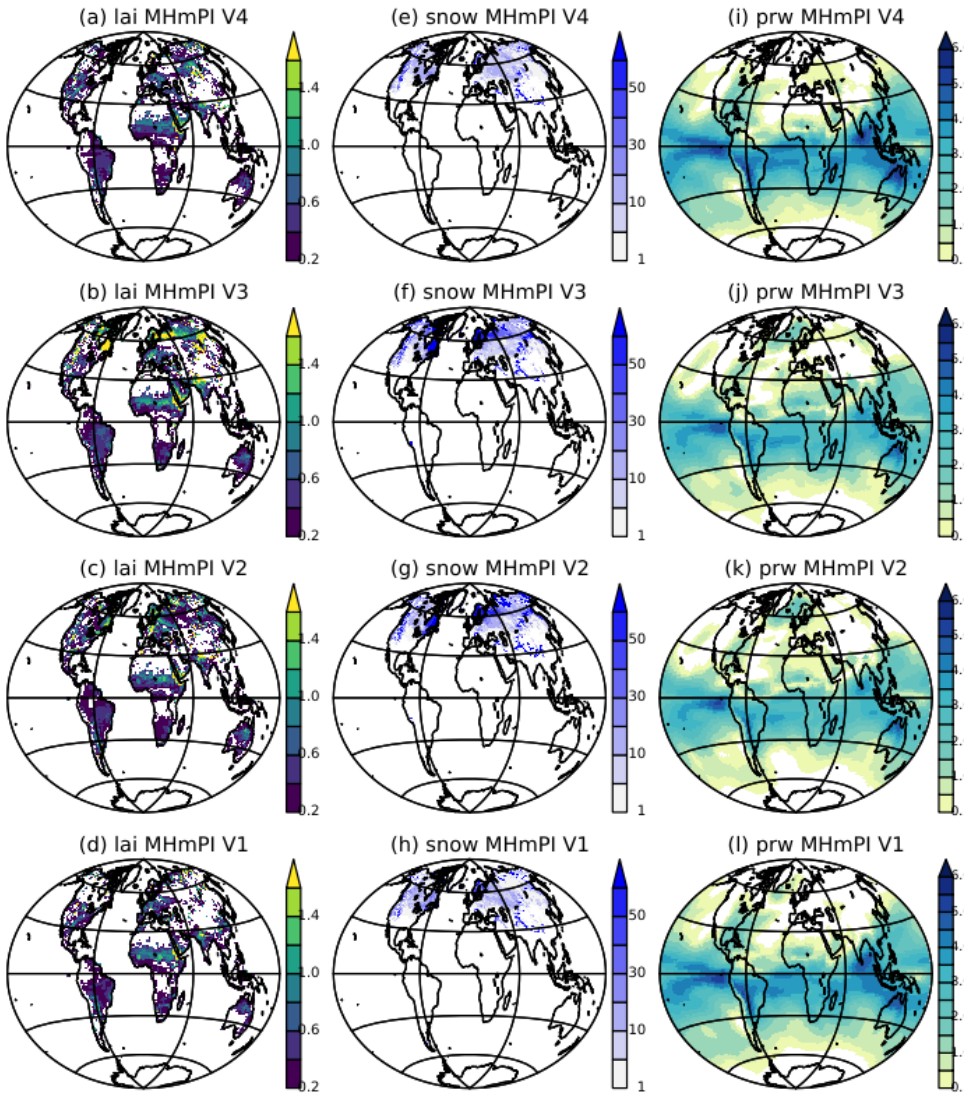

Figure 5: Root mean square difference between Mid Holocene (MH) and Preindustrial climates considering all combinations of 100-year annual mean cycles between the two periods at each grid points for (a) to (d) lai, (d) to (h) the snow mass (kg m$^{-2}$), and (i) to (l) the atmospheric water content (kg m$^{-2}$). Not that for snow mass the estimates have been restricted to 100-year monthly differences between February and May, which corresponds to the period where snow feedback over Eurasia is the largest between these two periods.



939

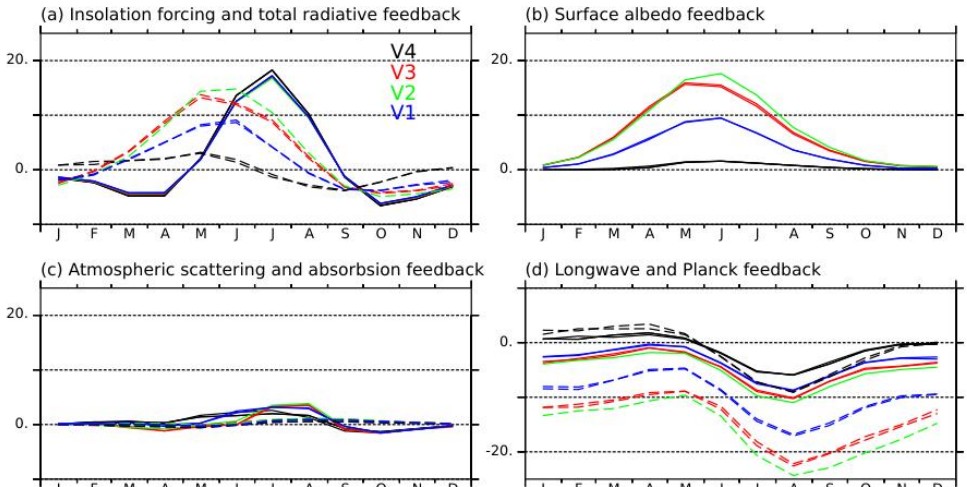

940

Figure 6: Radiative forcing and feebdacks estimated at Top of the atmosphere (W m$^{-2}$) between the mid-Holocene
and the predindustrial climates over the mid-to high latitudes in the Northern Hemisphere (45°N-80°N) for the
four model versions V1 (blue), V2 (green), V3 (red), and V4 (black). (a) Radiative forcing (solid lines) and total
radiative feedbacks (dash lines), (b) surface albedo feedback, (c) atmospheric scattering (solid lines) and
absorbsion (dash lines) feedbacks, and (d) longwave feedback (solid line) and Plack response (dash lines).

946



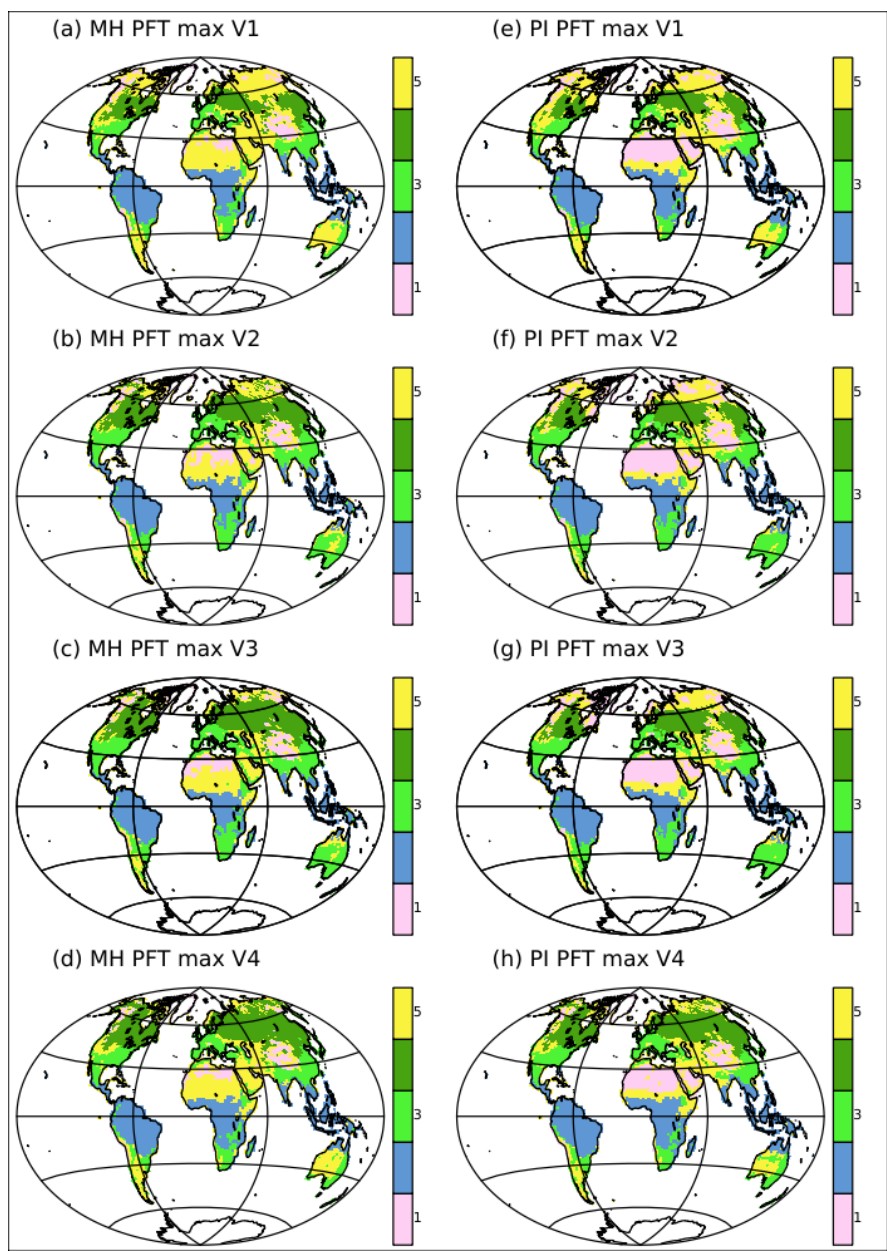

Figure 7. Dominant type of vegetation as simulations by the four model versions for (a) to (d) the mid-Holocene (MH) and ((d) to (f) the preindustrial (PI) climates. For clarity the 15 plant functional types (pft) have been groups into 5 major vegetation types: 1, bare soil; 2. Tropical forest, 3. Temperate forest, 4. Boreal forest, and 5 grass. These maps represent the vegetation average over the length of the simulation, without considering the first 300 year.



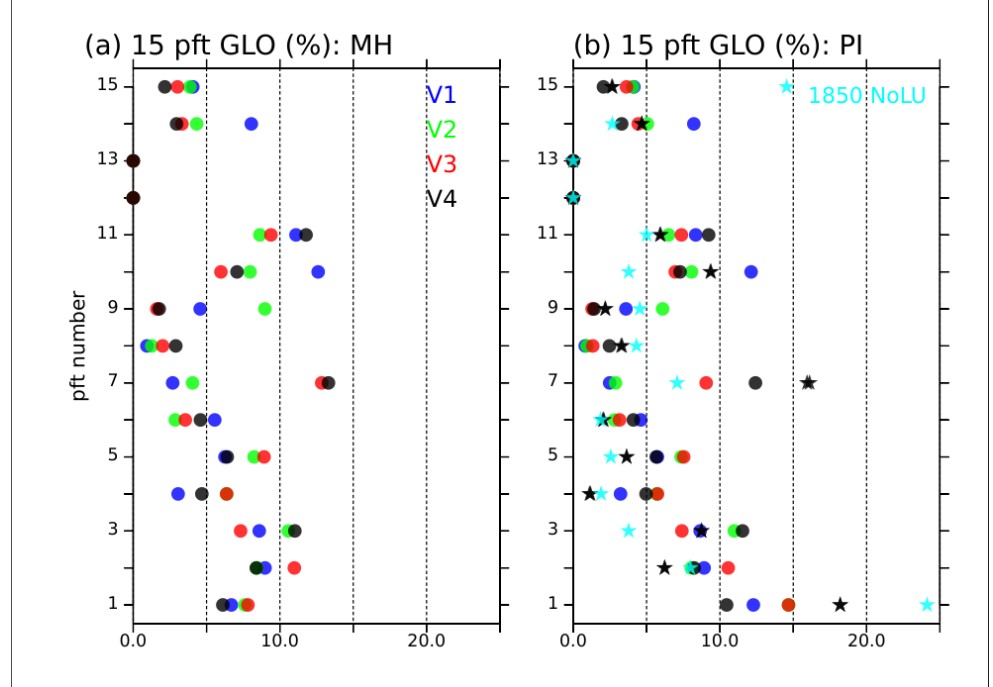

Figure 8. Percentage of global land surface covered by the different types of vegetation (pft) for (a) the mid-Holocene and (b) the preindustrial climates and the four model version (V1: blue, V2: green, V3: red and V4: black). The numbers on the vertical axis refer to the different pft, with 1 for baresoil, 2 for Tropical Broadleaf Evergreen, 3 for Tropical Broadleaf Raingreen, 4 for Temperate Needleleaf Evergreen, 5 for Temperate Broadleaf Evergreen, 6 for Temreferperate Broadleaf Summergreen, 7 for Boreal Needleleaf Evergreen, 8 for Boreal Broadleaf Summergreen, 9 for Boreal Needleleaf Deciduous, 10 for Temperate Natural Grassland (C3), 11 for Natural Grassland (C4), 12 and 13 for crops C3 and crops C4 that are not considered in this study, 14 for Tropical Natural Grassland (C3), and 15 for Boreal Natural Grassland (C3). The stars in (b) represent the pft distribution when only grid points that are not affected by land use in the observed 1850 vegetation map used as reference in simulations with prescribed vegetation, so as to compare the simulated natural vegetation with observations (turqoise for observations, and black for V4).



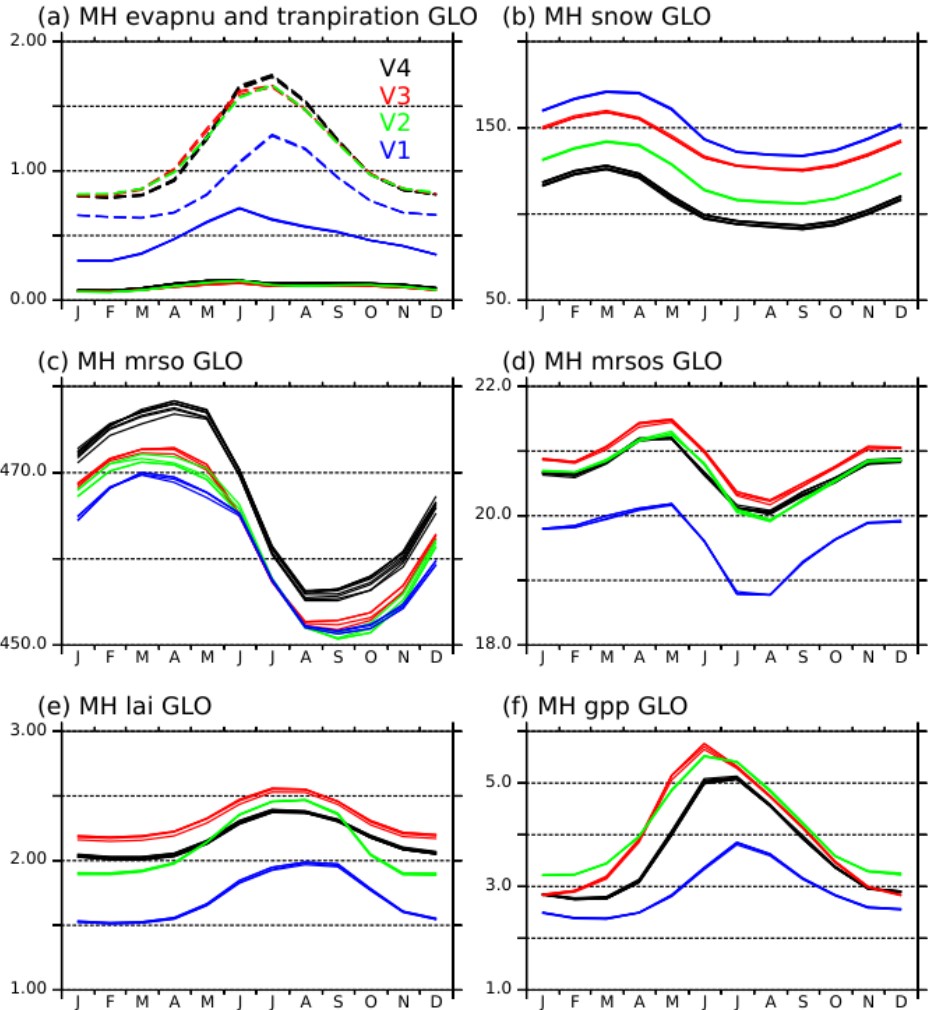

969

Figure 9. Annual cycle of mid-Holocene (a) bare soil evaporation (mm d$^{-1}$, solid lines) and transpiration (mm d$^{-1}$,
dash lines), (b) snow mass (kg m$^{-2}$), (c) total soil moisture (kg m$^{-2}$), (d) surface soil moisture (kg m$^{-2}$), (e) leaf area
index (lai) and (f) net assimilation of carbon by the vegetation (gpp, gC m$^{-2}$ s$^{-1}$) globally averaged over land for
the 4 simulations (V1: blue, V2 : green, V3: red, and V4 : black). All 100 years annual mean cycles, excluding
the 300 first years, are plotted for each simulation in order to provide and idea of 100-year variability and show
that the differences between the simulations are robust.

976

977



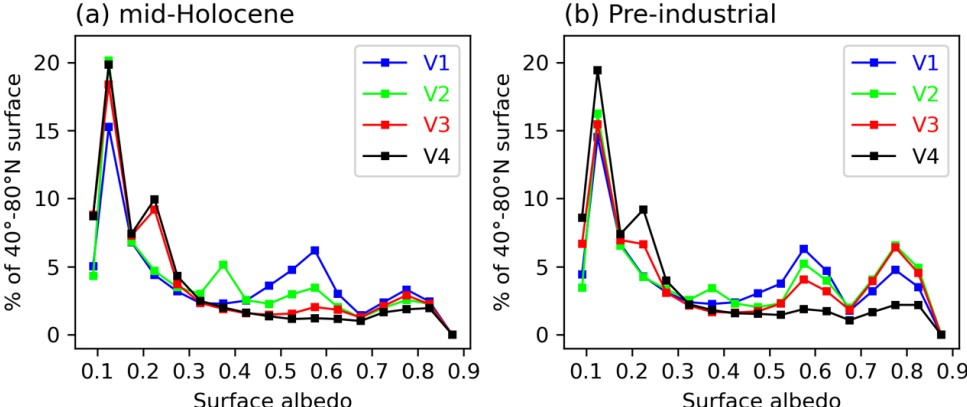

Figure 10: Distribution of the surface albedo (fraction of reflected radiation) simulations with the different version of the IPSL model, considering all grid points between 45°N and 80°N and months, for (a) the mid-Holocene and (b) the preindustrial climates. For each albedo bin, the value represents the percentage of surface with this particular albedo value. The first bin (lower value) corresponds to the ocean portion in the considered grid box where ice or land is also present. The higher values correspond to sea ice whereas values between 0.1 and 0.3 correspond to vegetation and bare soil, and values between 0.3 and 0.7 to different mixtures of vegetation and snow albedo. The surface albedo has been estimated using the surface upward and downward solar radiation.

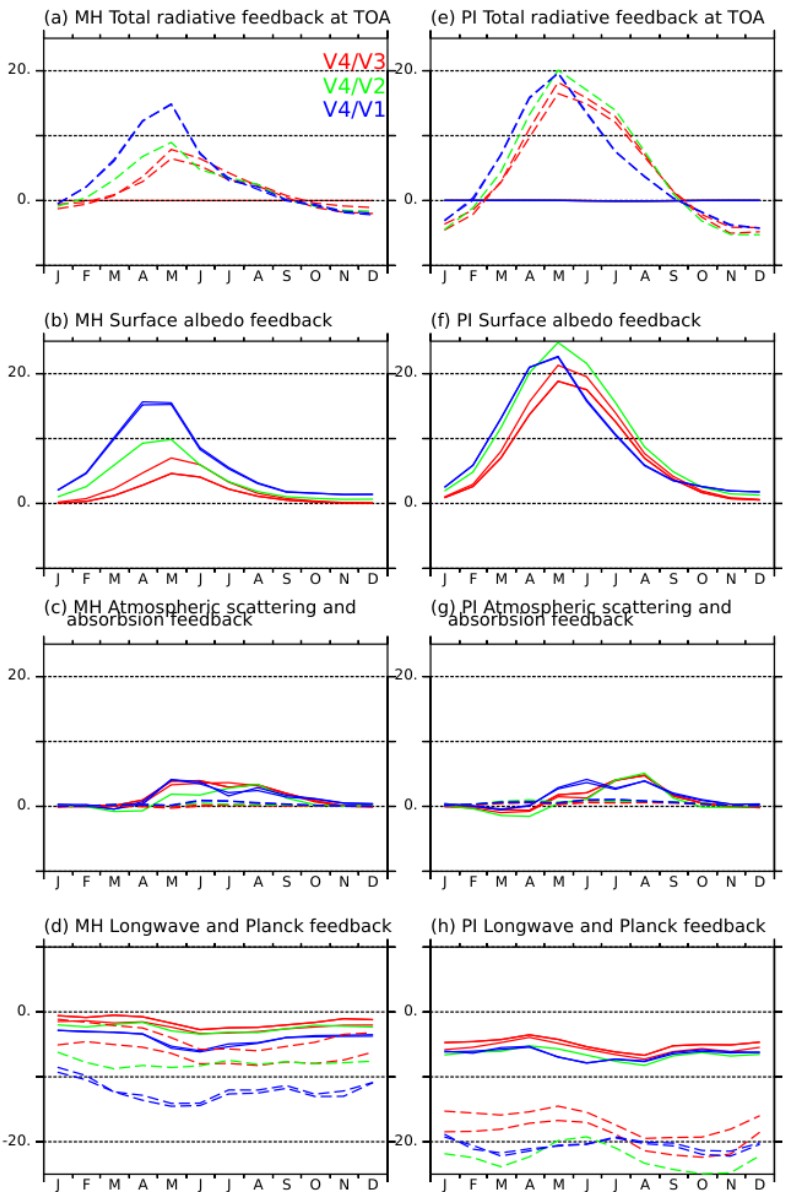

Figure 11. Estimation of radiative feedbacks (W m⁻²) induced by the differences in the land surface model and
vegetation between the V1 (blue), the V2 (green), the V3 (red) and the V4 model versions used as reference, for
(a) to (d) the mid-Holocene simulations (MH) and (e) to ((h) the preindustrial simulations. Positive (negative)
value indicate that more energy is entering (leaving) the climate system in V4 compared to the other version at the
top of the atmosphere.  As in figure 7 the different panels consider (a) and (e) the total radiative feedbacks, (b) and
(f) the surface albedo feedback, (c) and (g) the atmospheric scattering (solid lines) and absorption (dash lines), and
(d) and (h) the outgoing longwave radiation feedback (solid lines) and the Planck response (dash lines).





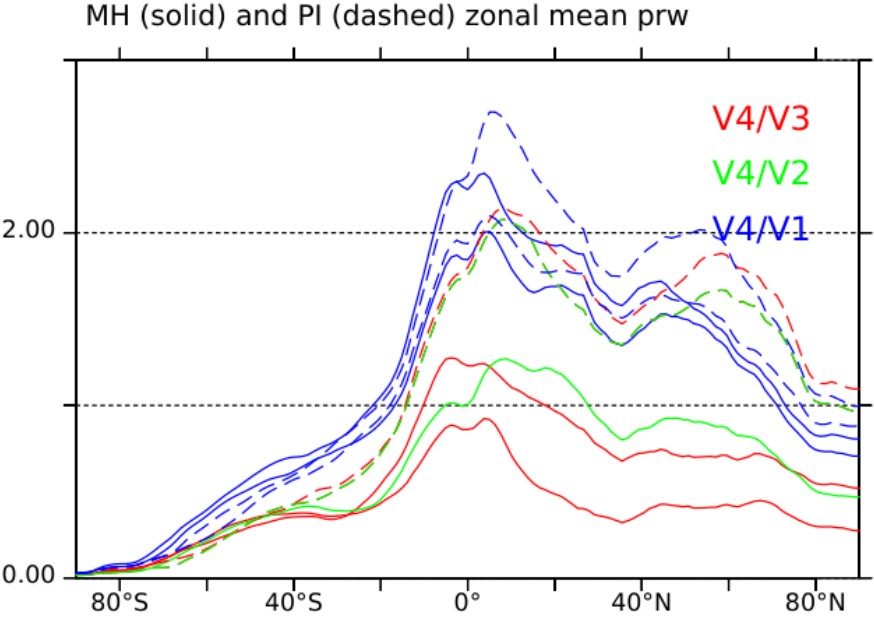

Figure 12: Difference between the zonal average of the integrated atmospheric water content (kg m$^{-2}$) as simulated using the V4 version of the model and the V1 version (blue), the V2 version (green) and the V3 version (red). The solid lines stand for the differences computed for the mid-Holocene climate and the dotted lines for the preindustrial climate. The different lines for a given estimate represent uncertainties computed using estimates from two different 100-year averages between simulations.





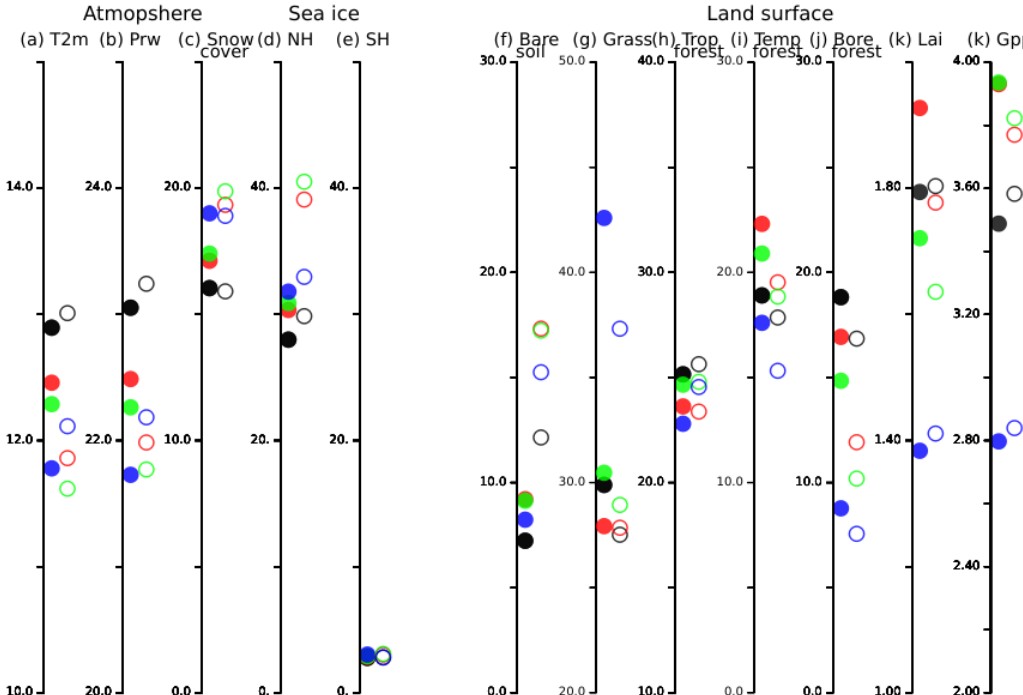


Figure 13. Mid-Holocene (full circles) and Preindustrial (circle) global annual mean for (a) surface air temperature
(T2m, °C), (b) Precipitable water content (kg m$^{-2}$), (c) snow cover over land (%), (d) bare soil (%), (e) grass (%),
(f) tropical forest (%), (g) temperate forest (%), (h) boreal forest (%), (i) lai, and  (j) gross primary production
(10$^5$ gC m$^{-2}$ s$^{-1}$)  and the four model versions (V1 : blue, V2: green, V3 : red and V4 : black).



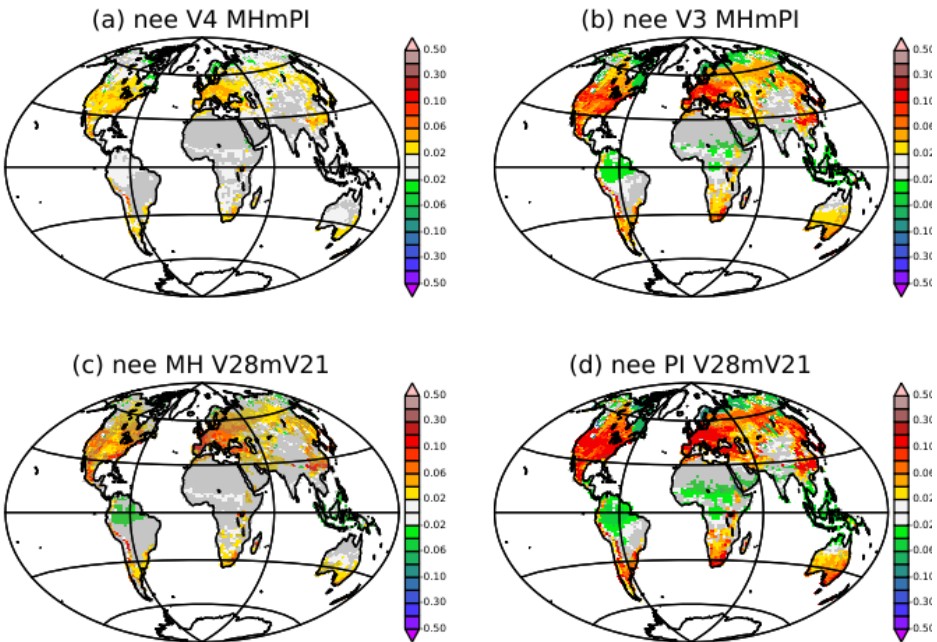

Figure 14. Net carbon flux from the vegetation (kg m$^{-2}$ s) difference between (a) the mid-Holocene and the preindustrial climates as simulated with version V4, (b) the mid-Holocene and the preindustrial climate as simulated by with version V3, (c) the version V4 and V3 of the mid-Holocene simulations, and (d) the version V4 and V3 of the preindustrial climate. Changes are considered to be significant at the 5% level outside the grey zones. The significance is estimated from all combinations of differences between 100-year averages between the simulations considered in each panel. For these estimates the first 300 years of the simulations are excluded.