# Peer review of "Dynamic vegetation highlights first-order climate feedbacks and their dependence on the climate mean state"

_EGUsphere, 2024_

## Author Comment (AC1)

We would like first to thank the reviewer for the useful comments. Several important points have been raised that will be considered for the revision of the manuscript.

*I recommend publication of the study with minor corrections, but I strongly suggest skipping the term "unavoidable" in the title and in the text. This term is not explained in the text. It just sounds alarmistic, as if the model would have a choice to "avoid" any negative consequences of model tuning. Quite the contrary, model tuning is done to improve the model performance.*

The term was indeed chosen to call attention to these feedbacks that are difficult to anticipate and understand in a fully coupled system. However, the two reviewers have a different interpretation of the use of this term compared to our intention. We therefore propose replacing it with 'first order', as this study is indeed discussing these well-known but poorly understood first-order coupled feedbacks. We propose to change the title to :

"Dynamic vegetation highlights first-order climate feedbacks and their dependence on the climate mean state "

*When reading the paper, I see that the new IPSLCM6 yields a much greener mid-Holocene Sahara than the former IPSLCM5 did. That is an exciting result. The authors highlight this achievement in one sentence (line 278/9) and a half-sentence (line 284/285). It would deserve more appreciation in the conclusions. Perhaps, a figure with a zoom on African biomes, using the biomization tool by Dallmeyer et al (2019), for example, which has already been applied to ORCIDEE PFTs, would be useful for a better comparison with other ESM simulations. But I leave this to the authors to decide.*

We haven't analysed the results in Africa sufficiently to be able to provide a well-grounded explanation of the different aspects that lead to the representation of the green Sahara in this version of the IPSL model. This should be the focus of another publication that we are planning on this topic. We are also very interested to understand better why the new version of the model represents the African humid period. We will thus only note that this was not anticipated from our mid-Holocene PMIP simulations (Braconnot et al. 2021), and we will suppress the end of the paragraph (lines 283 to 285).

*Perhaps a more formal analysis using the Alpert-Stein factor separation would yield a better understanding of feedbacks and synergies between feedback. But, again, I do not insist on doing a new analysis which would require $2^4$ simulations. It would be sufficient to mention that the present study does not differentiate between the pure contributions triggered by a new parameterization and the possible synergies emerging from combining new parameterizations.*

We agree with the reviewer that this would be a good idea. However, the traditional method of performing such a decomposition using coupled and standalone simulations is not entirely appropriate for high latitudes. This is because sea ice is modelled differently in atmosphere-alone simulations and in the fully coupled system, which affects the snow-ice albedo feedback and introduces errors in the estimates. In this manuscript, we focus on the total effect of the changes made to the model in the different versions. We emphasised the separation of

atmospheric feedbacks. This is why the choice was to use the simplified partial radiative perturbation approach to disentangle the role of atmospheric and surface albedo feedback on the atmospheric radiative budget. In the revised version of the manuscript, we will ensure that the fact that we do not estimate vegetation feedback itself is clearly stated.

*Finally, I suggest skipping trivial common places like the very last paragraph (lines 630 – 634). It is completely true that dynamical vegetation is an important factor which should be considered in ESMs. But this study is not the first one to point to the importance of dynamic vegetation. We (including the authors) have convincingly addressed this topic by numerous studies over the last roughly 30 years.*

This point is well noted. We fully agree that we are not the first to show that dynamical vegetation is an important factor. We will suppress or reformulate the text where needed to clarify the specific points we address in this manuscript. There are few studies connecting feedbacks to global energetics and energy conservation in a fully coupled model, which is what we do here. The two reviewers have made comments on the sentence in which we refer to global conservation, which suggests that this aspect is not entirely clear. We will expand this sentence.

*Minor comments:*

*Line 116 and following: It would be useful to learn something about the interaction with the C-cycle. Into which carbon pools of the plants and the soil is the carbon gain by photosynthesis fed? Or does this issue do not play any role here?*

This certainly plays a role, but there is no change between the model versions. We will provide a brief description here and add a comment later in the text explaining that an increase in GPP first leads to an increase in NPP, and then to an increase in biomass. This affects the carbon in the soil. In cold boreal forests, where decomposition is slow, a small increase in GPP induces a large relative increase in soil carbon.

*Line 242, Fig. 3: The abbreviations in the title lines (dtas, dpr) are not defined in the caption. Why not put a \Delta T_s or \Delta P_r in the title lines and in the caption?*

To avoid too long titles, we use tas and pr because these are standard CMIP names. We will reconsider it.

*Line 257, Fig. 4: The new parameterizations increase the simulated annual mean precipitation in WA, but still, the simulated precip amounts to only a factor of 0.4 of the reconstructed precip. How is the aggregation of data points and comparison with grid box results done? Using any area-mean? (Would be sensible to only consider grid boxes for which reconstructions are available.)*

Thank you for this remark. A short paragraph explaining how we did it disappeared between the different versions of the manuscript and we didn't realise. The model sampling is indeed done where there are reconstructions. This follows what was done in Braconnot et al. 2021. We will complete the explanation in the text and add a sentence to the figure caption.

*Line 283 ff: "It results from vegetation feedbacks amplified by synergy with ocean feedbacks …" surely, it does. But without differentiation between feedback and synergy, it remains a*

*trivial statement and could be skipped – in contrast to the second half of the sentence which likely is the real reason and would deserve more attention.*

Well noted

*Line 287, Fig. 5: The labels on the colorbars are partly hidden behind the colorbars. Please shift. The global maps, specifically for the differences in lai, are too small to see any details outside the tropics. Please enlarge the figures to the size of the other global maps in the other figures.*

We will fix this, there is a bug somewhere…

*Line 304: It would be helpful to note that alpha_p is the surface albedo. Commonly, one would symbolize the planetary albedo with the subscript 'p'.*

Alpha _p is the planetary abedo and alpha_s the surface albedo, this is provided line 310

We will add alpha_s in parenthesis on line 304 to avoid any confusion.

Eq.(3) and other places in the text: Sometimes the subscripts appear as subscripts, sometimes as an extension of the variable, for example as in SWsi  vs. SW_{si}  or LWsup vs. LW_{sup}. Please harmonize.

Well noted. It will be done

Line 318: gases instead of gazes

Well noted, thank you

*Line 348: What are pft 7 and 8? It would help reading, if the names of the PFTs are mentioned here or in a table.*

Line 454 and other places: Sahara Sahel or Sahel Sahara sounds a bit cumbersome, because the Sahara and the Sahel (region) are pretty different regions.

We will clarify this

*Line 457: …, so that the magnitude …*

Noted

*Line 509: "The suite of mid-Holocene  … allow us to dig into the complexity of the Earth's climate system." That is a rather generic and bold statement as this study just touches a small subset and very specific aspects of the global climate system.*

We will rephrase to be more precise here.

*Line 510: "We insist on the fact …" I do not understand, why you have to 'insist' on the fact, instead of highlighting the fact.*

This is a word that does not have exactly the same meaning in French in this context. We will revisit the sentence.

*Line 519 ff: I do not quite understand the meaning of this sentence. Perhaps it is just the wording 'associate to' … The word 'fulfil' should be 'fulfill'.*

We will clarify

*Line 527: "We show that dynamical vegetation reveals how …." I am not sure how dynamical vegetation can reveal anything. The analysis of the climate-vegetation interaction can certainly do, but, again, only with respect to the processes considered, not with respect of the entire complexity of all biospheric processes.*

We agree. We will rephrase

*Line 537: Which "step changes between the model version … is (shouldn't is 'are') different from …?*

We will add precision

*Line 542: Which "model content" … lead(s) to different vegetation cover …?*

The physics of all components except the land surface is the same between model versions and constraints model feedbacks. This is where we will provide more detail on aspects of the studies that depend on the model and those that are more generic, in order to address the point raised at the beginning of the review.

*Line 562 ff. Indeed, the statement that "simulated vegetation is an integrator …" is "trivial". Perhaps a more modest statement would be sensible. This study is not the very first one to highlight the importance of vegetation dynamics.*

We will consider it.

*Line 565/567. I agree that one cannot infer vegetation feedbacks from studies in which vegetation patterns are kept fixed. In this sense, the titles of early studies (e.g. Kutzbach et al. Nature 1996) are misleading. These studies analyzed impacts rather than feedbacks.*

We will consider the remark and complete the text

*Line 596: This would require (instead of requires)*

Noted

*Line 600: … because land use (not land used) is not*

For sure, thank you.

---

## Author Comment (AC2)

**Response to the comments made by reviewer 2**

We would like to thank the reviewer for the careful reading of the document and the useful comments, which will help us to improve the manuscript.

1) *1 The main finding is that vegetation feedbacks are incredibly important and that they vary seasonally and by location. These have been quantified for 4 different model configurations. From this I'm left wondering whether this is a way to discern more clearly between the model versions. Even if this is not done here, it could be discussed what observations from either present-day or from the mid-Holocene would be required.*

This is an important comment. Yes, the dynamic vegetation is a way to distinguish more clearly between different model versions. Indeed, it provides indications of critical aspects to look at in a fully coupled system, such as the soil evaporation in spring in mid-latitudes or how the photosynthesis parametrisation triggers the plant seasonal development. While this is not a new concept, this study introduces new elements that can inform Earth system thinking. This way of thinking is still difficult to incorporate into model development, as most approaches still rely on impact-based reasoning rather than feedback-based reasoning. The mechanisms we discuss and the way they trigger atmospheric feedbacks are model-independent. What is model-dependent is the mean climate state, which depends on these factors and, critically, on the atmospheric or ocean-ice physics (for the timescales considered here).  We will reinforce the conclusion on these aspects. As requested, we will also expand the discussion about observations, without adding model-data diagnoses to this manuscript. However, the key point we raise in the conclusion is that it is almost impossible to find the right way to evaluate the model. Present-day observations are affected by land use, and paleoclimate data are indirect. We can address this by examining different types of paleoclimate indicators. More importantly, however, the available reconstructions still have incomplete data coverage in several key regions. This is why, in the conclusion, we propose that looking at different past periods for which changes in seasonality are the dominant factor. Together with the preindustrial and the present-day climates, these past periods, such as the mid-Holocene or the Eemian, allow us to evaluate the ability of a climate model to reproduce seasonality and the seasonal feedbacks, considering both the seasonal processes and the factors arising from differences in the climate mean state.

2) *The Discussion and Conclusion section lacks focus and covers perhaps too many topics. I think this paper would have much greater impact if these two aspects could be separated and a more concise and clear Conclusions section were to be developed.*

We agree that the discussion and conclusion sections are too long and should be reorganised and refocused. We have two possible solutions for it. The first one is to keep the discussion and conclusion section and add subtitles. The second solution is the one proposed by the reviewer, which consists in adding a Section 5 before the conclusion. We propose to do this and add a section 5 "Synthesis and implication for the carbon fluxes," where we will discuss fig 13 and 14. The conclusion, section 6, will then summarise the key findings and provide a perspective for model evaluation and model development.

3) *Editing for grammar, typos and figure presentation is needed.*

Thank you for highlighting these typos and errors. Some of these typos could have been avoided. Others are more complex to detect for non-native English, and have not been detected by our English corrector. We will improve this. Regarding the figures, we will consider the different remarks and adjust the figures accordingly.  Concerning the cropped edges of figures, the small piece missing for one of them results from the inclusion of the figures in Word tables for the production of the complete

manuscript. The original figures are correct. The issue with the numbers in the legend colour bars it is more complex, and requires fixing a bug to improve this. As suggested, we will first try to reduce the number of colour bars so as to enlarge the size of the maps..

**Minor corrections:**

*Title: I find the word "unavoidable" slightly misleading here. It has connotations of committed climate change etc. I recommend rewording throughout with something like robust or parameterisation-independent.*

We understand this comment. It is consistent with Reviewer 1's comment. The term was chosen to draw attention to the fact that these feedbacks are difficult to anticipate and understand in a fully coupled system. Since our discussion focuses on first-order feedback, we will replace the term with 'first-order', which is consistent with the way we discuss these feedbacks in the manuscript.

We propose therefore changing the title to :

"Dynamic vegetation highlights first-order climate feedbacks and their dependence on the climate mean state "

*Line 132: This is very similar to the changes to soil moistures stress in transient Holocene simulations by Hopcroft & Valdes, 2021 PNAS.*

Our changes consist of adding biomass-dependent resistance to bare soil evaporation. As far as we understand, it is not exactly the same as in Hopcroft and Valdes (2021, PNAS). The changes they made in their study affect all PFTs and thus have a major impact everywhere. There is already soil moisture stress for the different PFTs in ORCHIDEE, and we have kept this as it is. Here, we only consider bare soil moisture stress, and thus the proportion of total evaporation between plants and soil. This explains why the effect is significant in mid-latitudes and in spring, subsequently affecting tree growth, whereas it is small in the Sahel or has almost no effect on the green Sahara. Although we are specialists in the African monsoon, we have chosen not to discuss Africa in too much detail in this manuscript, focusing instead on the differences between the simulations.

*Figure 4: The cyan (data) points are not easy to see. Could you redraw using thicker lines for the data points?**

Yes, we will improve this, and also add the large error bars for the reconstructions (they are considered in Braconnot et al. 2021).

*Figure 7: for clarity could you consider creating a single colour bar for all panels and labelling it with PFT groupings instead of numbers.*

Yes ,we will try to do it if we find the way to adjust the relative size of the panels.

*Figure 8: similar comment as above - label the y-axis with the PFT names not numbers.*

This is more difficult to do. The reason is that the PFT names are too long and would appear too small.  We will certainly use PFT acronyms instead.

*Line 260: "We synthesize the mid-Holocene differences with preindustrial by showing the mean root mean square difference between the two climates in Fig. 5 for leaf area index (lai), snow, and atmospheric water content."*

*It's not clear why this choice is made at this point. It will compress everything to be a positive anomaly which is reducing the information. Is this intended?*

Yes, it is. The annual mean is the residual of large seasonal variations. This is a way of showing on a map where the largest variations occurred between the mid-Holocene and the pre-industrial period, taking into account shifts in the annual mean and changes in seasonality (magnitude and seasonal phase).

*Lune 278: "all of these model versions produce a green Sahara"*

*I'm not sure I agree with this. The precipitation anomaly shown in figure 4 is too small and the LAI anomaly is only covering half of the Sahara?*

Yes, they do, and grass is the dominant PFT. Note that we use a threshold of 0.2 for the LAI map; thus, there is a visual artefact. We will revisit the figure, trying to use a different colour map and, as already suggested, suppressing the redundant colour legend.

*Line 284: "and from atmospheric physics and land surface improvement between the IPSLCM5 and IPSLCM6 versions of the IPSL model (Boucher et al., 2020; Hourdin et al., 2020)."*

*I'm not sure this is very well supported. can you either explain in more detail ?*

You are right; it is not well supported, so we will suppress this sentence. It is based on the authors' knowledge of the model and still needs to be fully analysed. This is beyond the scope of this manuscript.

*Lines 333-336: "The snow albedo effect is amplified*

*334 when grass is replaced by forest in the mid-Holocene simulation, which occurs over a large area in Eurasia for V2*

*335 and V3 compared to V1 where grass is dominant or V4 where a larger fraction of forest is still present in the*

*336 preindustrial simulation (Fig. 7)."*

*Should this be the other way around or could you clarify? Grass being replaced with trees would result in lower albedo overall because trees are lower albedo than grasses and trees cannot be covered as efficiently by snow as can grass?*

The concept of work amplification applies to both positive and negative effects. This sentence seems to cause some confusion. We will rephrase it for clarity.

*Line 362: "It appears to be a critical model aspect contributing to a better representation of boreal forest."*

*Again I'm not sure I agree as the difference in the boreal forest pft 7 seems very small between V1 and V2.*

PFT 7 is just one of the boreal forest PFTs. The boreal forest encompasses PFTs 7 to 9. PFT 9 covers a large area in V2, but the cold climate still prevents PFT 7 and 8 from expanding. So yes, it is a critical aspect, and all the tests we have conducted confirm that it is particularly critical for mid latitudes. We have already adjusted bare soil evaporation by a factor in the previous simulations with dynamical vegetation (a had-hoc solution). This was already done to limit evaporation in spring and allow vegetation to grow in the mid-latitudes.

**Technical corrections**

*Overall there are a lot of minor typos, grammatical errors and cropped edges of figures. Some of these are included below.*

Thank you for highlighting the typos and remaining errors. regarding the figures, the issue with the numbers in the legend is a bug that needs to fixed. We will revisit the layout of the concerned figures by reducing the number of colour legends.

*Line 9: "with the IPSL climate models for which dynamic vegetation is switch on." This should be: switched on*

Thank you

*Lines 16-17: "which are needed to fulfill the global energy conservation constraint of the climate system."*

*I don't really understand what this means in this context.*

This is an important point. These are coupled equilibrium experiments, for which energy conservation is a strong model constraint. We will rephrase this and provide the missing explanations in the text.

*Lines 18:control -> controls*

*Line 18:nb"Photosynthesis parameterization .." should be "The photosynthesis parameterization ..."*

*Line 25:"The Green Sahara"*

*Line 41: "The increase \*in the\* number of "*

*Line 42: "has emphasize"  -> "has emphasized"*

Thank you for highlighting these errors. We will correct them and improve the way English is checked throughout the document.

*Line 129-131: "This adjustment in the bare soil evaporation parameterization was not incorporated into IPSLCM6A-LR due to the fact that it induces a surface warming that was*

*not fully understood to be used in the whole suite of CMIP6 simulations (Cheruy et al., 2020)."*

*This is a grammatical error in this sentence.*

We propose to supress the end of the sentence after understood.

*Line 896: "the vcmax curves are plotted toe a mean temperature"typo*

*Line 152-155: "Another important difference is that in PhotoCM6, the response to temperature is adapted to the local long term (i.e. 10 years) temperature of each pixel whereas in PhotoCM6, the temperature dependence is fixed for the whole pft."*

*This does not make sense to me.*

The second one should definitely be PhotoCM5. We will revisit this sentence. It's a correction that should have been made before the submission.

*Line 198: "It guaranties the entire consistency between the simulated climate and the simulated vegetation."*

*This doesn't really make sense to me.*

This is the only way to ensure that all parameters used in the land surface model are reinitialised and consistent across the model's dynamics, hydrology, and carbon components. We can suppress this sentence as it refers to minor inconsistencies that have been corrected depending on whether the model is used offline, online, with or without dynamical vegetation. Therefore, the reference to Braconnot et al. (2019) in the previous sentence is sufficient.

*Line 217: "A conclusion from Fig. 1 is that 300 years of"        this should be figure 2.*

 Yes, you are right. We added figure 1 late in the writing process and forgot to update this number

*Line 244: |standard IPSL model without dynamical vegetation"which model configuration is that  - state here please.*

We will adjust this sentence. It refers to the PMIP4 mid-Holocene simulations that were run using the IPSLCM6 model (Braconnot et al., 2021).

*Line 302: "atmospheric diffusion do you mean scattering and absorption?*

Yes, this is an error. We will correct

*Line 296: "Positive values (negative) indicate that the feedback brings more (less) energy to the climate system in V4"*

*These double meaning sentences in brackets are in my opinion extremely hard to read and should be avoided. e.g. https://doi.org/10.1029/2010EO450004*

We agree and will suppress the text in brackets here.

*Line 510-511: "We insist on the fact that climate-vegetation interactions induce seasonal feed backs that trigger unavoidable first order albedo and water vapor radiative feedbacks"*

*This use of "insist" and "unavoidable" comes across a little odd. Could you clarify e.g. "We find" instead of we insist, and instead of unavoidable use a word like robust or parameterisation-independent?*

This comment is similar to a comment by reviewer 1. We will revisit the sentence

*Lines 517-521: "The LW radiative feedback is less discussed when the role of vegetation is inferred from vegetation alone simulations or simulations where the sea surface temperature and sea-ice cover are prescribed. It is a first order effect associate to the change in temperature and fulfil the convective radiative equilibrium which serves as a basis for the reasoning on climate sensitivity (Dufresne and Bony, 2008; Manabe and Wetherald, 1975; Sherwood et al., 2020)"*

*This isn't clear at all.*

We will detail a little bit here coming back to the constraint on the global energy conservation in the ESM model, and the long wave and short-wave balance needed at equilibrium. Intermediate explanations are indeed needed, including the fact that the radiative balance can be broken in atmosphere alone simulations.

*Figure 4: it's really not clear which circle is what in this figure. Please improve the legend.*

We will do it.

*Line 934" Not that" should be "Note that"*

Thank you

*FIgure 9: make panel titles in English not in model variable codenames please. e.g. total soil moisture instead of mrso.*

We agree. We'll have to find a way to keep this long name readable.

*Figure 13: consider connecting the same-coloured dots with lines for clarity?*

We will not do it because we already tested it and we know the figure becomes a mess.

*Figure 14: this and other figures have edges of the figure cropped.*

For this figure, it comes from the way the figure was included in a table in the Word file. The original figure is correct.

---

## Author Response (AR2)

**Response to reviewer's two comments.**

We would like first to thank the reviewer for the careful reading of the manuscript and the useful and constructive comments. Indeed, we found these comments useful to understand why and where some of the difficult aspects of the paper would benefit from clarification. In the following, we include the reviewer's comments in italic and blue, and our response below each major comment in black.

**General comments**

With the interdisciplinary readership of ESD in mind, I recommend expanding more on the study's motivation and context, especially in the abstract and introduction. Currently, the manuscript lacks context/explanation at the beginning of both the abstract and the introduction. As a result, the entry barrier to the manuscript could be very high for readers who are not already familiar with the thinking of land modelers. Beyond that, the manuscript takes it somewhat as a given that the chosen time frames (PI, MH) and parametrizations (bare soil, photosynthesis, tcrit) are of interest. The same goes for introducing the relevance of vegetation feedbacks and the reasoning behind DGVMs as a whole. I reckon re-ordering thoughts popping up here and there in the introduction could reduce these entry barriers and improve its readability at the same time. For example, I would recommend touching on the PI and MH time frames much earlier - right now, this is done in greater detail only at the end of the introduction, but it would be helpful a lot earlier. Another example would be to touch on vegetation feedback with concrete examples early on, e.g., after the first sentence of the introduction, rather than jumping right to the standard literature review.

Following the reviewer's comments and suggestions, we reordered the introduction and added some context upfront. We also revisited the abstract accordingly.

In addition to that, I suggest the authors expand on how they employ the term "feedback" throughout the manuscript. On the one hand, they modify the strength and character of the vegetation's response to climate, which alters the vegetation-climate feedback loops, which is clearly formulated. However, on the other hand, they compare the radiative effects between the different model configurations/climate states and term the result of this analysis "feedback" as well (e.g. Figures 6, 11). Although one can "sense" the proximity of these radiative effects to the concept of a feedback factor (think ECS as an example), their understanding is not precisely described. As a result, I am asking myself the question to what degree the difference in radiative effect of say the "surface albedo feedback" diagnosed between the two climate states (Fig 6) is directly comparable to the radiative effect diagnosed between different model versions (Fig 11). Thus, I am missing a brief explanation of what is considered the forcing in the different analyses. Maybe adding a conceptual figure to the introduction or to Section 3.3 would have avoided my confusion.

During the first round of revisions, we made the effort to call radiative feedbacks the feedbacks connected to the estimation of climate sensitivity, and use the generic word "feedback" for vegetation-climate interactions. There are indeed some places where this was not fully done. We also agree that explaining it quite early in the manuscript would be useful, and we added it in the introduction. In some cases, the word "feedback" was replaced by "interactions" when dealing with vegetation-climate interactions rather than directly addressing vegetation-climate feedbacks. It should help to avoid confusion in the terminology. We also clarified why we use the same framework to diagnose radiative feedbacks between two climatic periods or for the same climate between two model versions. This is done both in the introduction and in section 4.1.

Finally, I am convinced that providing the final manuscript to a native-language proofreader and utilizing one of the many spell- and grammar-checkers would greatly help improve its readability and accessibility. Currently, it contains a lot of typos and convoluted sentences. I would not expect an extremely polished text, but as a reader, I found it very hard to get the author's point on a number of occasions, and it should be in the author's interest to make the paper easily accessible to the reader.

The paper has gone through spell and grammar checkers. We hope that having clarified some of the places where there was a lack of context or where the sentences were too complex provides the needed improvement in the language. It is difficult to do better with the tools and resources we have.

I am listing a number of technical comments at the end of my statement, but this list is not exhaustive.

**Specific comments**

- General remark on figures: Many figures use a color scheme that is not colorblind-safe. I am not expecting the authors to necessarily change this aspect, but I would recommend keeping this in mind for upcoming work. Also, with the current projection of map plots, I found it challenging to recognize details in the high latitudes — This is a bit unfortunate, since a substantial part of the results centers around exactly those areas.

We understand these comments. This whole set of figures has been time-consuming to draw, and we are doing it using versions of software available at the computing center to be close to the archive. Unfortunately, everything is not possible, and there is a bug in pyferret that prevents us from having a correct figure when rotating the reference latitude and longitude for the projection. We have worked on the layout as much as we could.

- The authors have updated the title. However, I find the newly added word "highlight" inconclusive. To my understanding, the authors utilize it in the sense of "modulating" or "amplifying" in the text as well (e.g., 1 498), and in my opinion, both of these terms would describe their intention more precisely.

Well, the first version of the title, "Dynamic vegetation reveals unavoidable climate feedbacks and their dependence on climate mean state," was closer to the exact focus of the study and the reason why we decided to write this paper. In the first round of reviews, we had strong comments on the title and decided to change it. The confusion comes from the fact that it was interpreted by the reviewers as if we pretended to be the first to advocate the need to consider dynamic vegetation in climate change experiments. This is not our aim. Our aim is the analysis we do of the close imbrications of first-order feedbacks, and the fact that these strong linkages trigger unavoidable feedbacks we do not really know how to deal with when developing climate models. These feedbacks are highlighted when dynamic vegetation is considered. This is why we chose the word "highlight" rather than "reveal" (although in French, "reveal" in the context we are using it is better) and revisited the title.

The proposition of the reviewer to adjust the title once more is interesting. We will not adopt it, because it puts too much emphasis on the dynamic vegetation, when the emphasis is on the imbrication of feedbacks and the fact that the change in vegetation is part of this first-order feedback. In our opinion, the title that best reflects a "pitch" of the paper content is

"Dynamic vegetation highlights unavoidable first-order climate feedbacks and their dependence on climate mean state."

We thus propose this new version of the title and would appreciate feedback on it, or a discussion with the editor about it.

- l 26: mid-Holocene and pre-industrial climate  $\rightarrow$  mid-Holocene and pre-industrial vegetation state? (I am not convinced the green Sahara and boreal forest are part of the climate state in the physical sense.)

In this case, this is indeed the case. The Green Sahara and boreal forest are part of the climate in a physical sense and characterise climate change. We rewrote the introduction, and this sentence has been revisited. It now only mentions vegetation.

- l65: I find this statement a bit too much black and white, since there is more granularity to discuss on how much vegetation dynamics are (not) considered. Maybe the authors can expand a bit on this statement/add a reference.

We reorganised the introduction so that the comment is now better connected with the other references. We also slightly modified the sentence so that it doesn't appear too black and white to readers who have been considering these couplings for a long time.

- 1194/195: What kind of inconsistencies and why are they not relevant to this study? It has mainly to do with the way the different types of vegetation, carbon, and water were reallocated to very small fractions of vegetation. These inconsistencies do not alter the results, and when corrected, the results remain similar, with a poor representation of boreal forest in V1. The corrections were also needed to guarantee that the model is the same when switching off or on the dynamical vegetation.
- Section 2.3 title: "Vegetation-climate equilibration"? We prefer the word adjustment here. We completed the title to explicitly say "Vegetation-climate adjustments to incoming solar radiation and atmospheric composition".
- Figure 3/Section 5.3: Is Bartlein et al. 2011 really the most up-to-date/comprehensive reconstruction that is suitable? Also, it is only pollen-based and does not use multiple proxies. Would newer reconstructions like Erb et al. 2022 (Clim Past, multi-proxy but no pollen cores) be an alternative, or are there other reasons that are not discussed here?

Bartlein et al. 2011 is indeed not the most up-to-date reconstruction, but it provides a consistent reconstruction of biophysical variables (coldest month, warmest month, and not a particular month) that are better suited to relate to the life cycle of vegetation than summer or winter temperature or a particular month. In addition, it provides uncertainties on the reconstructions, and the lead author knows very well the limitations of this product. There are now interesting new products, such as the one by Erb 2022, with data assimilation using climate models. The authors lack expertise on this product and on the way to properly use it for model-data comparisons. The Erb 2022 manuscript also mentioned several caveats in the reconstructions, part of which come from the multiproxy approach and that these proxies may reflect different aspects of the climate, some of which might be different from the original interpretation. There are lots of caveats in model-data comparison for the Holocene. The choice for this paper was to use a well-known product we also used in our 2021 publication discussing the PMIP Holocene simulations with the IPSLCM6 model. Doing something different from what we did before would require a specific study and in-depth intercomparison of the different products and the quantification of uncertainties. This will certainly happen in the new phase of the Paleoclimate Modelling Intercomparison Project.

- Section 3.2: I wonder if it has been tested whether this approach is sensitive to the length of the different simulations. Since they have different lengths, they result in a different number of 100-year-long slices, and I could imagine this to have consequences for the statistics.

The approach is sensitive to the length of the simulations, as is any statistical method with little sampling. We checked with the 1000-year-long simulations that it doesn't affect the results when we consider only 200 or 400 years of the simulation to compute the cross differences. The limitation in the number of degrees of freedom is the reason why we directly provide the results of each 100 years and do not try to estimate the mean 100-year average and error bars using a Student's t-test or another non-parametric method.

- 1306-308: I actually do not agree with how the figure is interpreted here. From Fig.5 it appears that the LAI in V2 is similar to V4 as well. And it seems to me that the snow cover changes are definitely the largest for V2 and V3.

The comment is for Eurasia and eastern North America, and V2 and V3 have indeed the largest changes we wanted to highlight. Thank you, we didn't realise the error in the first parenthesis. It is now corrected to V2 and V3, the same for LAI and snow.

- 1313-315: From Fig.5 it also appears to me that V2 and V3 produce the lowest changes in precipitable water in the tropics, not the largest. In the high latitudes, the opposite applies.

In this paragraph, the comment is only for high latitudes. We adjusted the text of the paragraph and added a comment on the fact that changes in precipitable water are larger in the tropics for the other two simulations.

- *l400: Figure 10 is not referenced, but I think it is discussed here.* We added the reference to figure 10.
- *l410: I am pretty sure this should be Fig 11, not 12, referenced here.*This is right. We changed the order of the figures, and it seems that there were still some inconsistencies in the revised version.
- Figure 12: As far as I can see, it is neither referenced nor explained anywhere. Figure 12 is reference at line 432, 444 and should have been also referenced at line 487. We added the reference there.
- *The third-to-last paragraph in the Results is hard to read and could be simplified.*We adjusted the paragraph, supressed the parentheses and adopted a more direct style.
- 1524: I do not agree with how the finding is described here. The seasonality (difference between the highest and lowest amplitude) is stronger for V2, but it is not for V3, which in turn is very similar to V4. V1 resembles V2 more than it resembles V4.

We adjusted the text by referring now to the peak LAI during boreal summer. However, V1 and V4 are parallel, and V2, even if it resembles V4, has a slightly larger seasonality. But we agree that because of the offset of V1 with the other simulations, it was not easy to follow.

- 1532-535: The authors argue that a seasonally lower GPP drives a higher soil moisture content. However, I cannot infer a seasonally higher moisture content in V4 from Figure 9, although I do see an overall offset. I would rather interpret this in a way that seasonally lower GPP reduces (evapo-)transpiration, leading to higher soil moisture.

The reviewer's interpretation is the same as our interpretation. We didn't realise that "implies" would be interpreted as a direct link. Therefore, we added the link with the evapotranspiration, which is better.

- 1535-536: I am not convinced whether one can actually call this aspect counterintuitive. To me, it rather appears to be a straightforward consequence of the different photosynthesis schemes.

This is true; "counterintuitive" comes from the fact that, with a given photosynthesis scheme, a simulation with a larger vegetation cover would certainly have a larger LAI and GPP, and this reasoning is often applied when comparing model results. When the connection is made with the photosynthesis parameterisation, as we do here, it becomes obvious that this is not what happens between our simulations and that it arises as the result of the different formulation of the photosynthesis parameterisations. We slightly expanded the reasoning in the text.

- Section 5.3: When discussing land use effects, it should be noted that they also induce non-local effects. Therefore, the "NOLU" reference values could (and likely will) be affected by land use as far as I can see. Non-local effects could be briefly mentioned here.

We expanded the text in this section and added two references, Smith et al. 2016 and Marquer et al. 2023, to better introduce the effect of land use and recent publications from LUMIP or the REVEAL pollen data set combined with modelling.

- Section 6: As already pointed out by other referees, the case for DGVMs has been made before — which does not imply that the relevance for including them in ESM simulations should not be mentioned here. However, to perhaps suggest an additional aspect for the conclusions here: To me, this study is a great example for "no model (configuration) suits all needs" — one configuration/model might be better suited to simulate a Sahara greening, while another one might result in lower climate biases in another region of interest (and in addition climate is not just mean climate). And this diversity stems from the fact that the Earth system is highly complex and dynamic. Maybe the authors are interested in taking up this aspect.

The point raised by the reviewer is interesting. Even though we share it, we decided not to go as far as proposed. We simply added a sentence in the last paragraph highlighting that models cannot be perfect and compromises need to be made.

Technical corrections

**Multiple occasions:**

- "Northern Hemisphere" (capital letters) done with a systematic search
- "pre-industrial", "mid-Holocene" (mind the dash) done with a systematic search
- "fully coupled/fully-coupled" (adopt one convention)
  We are using fully coupled throughout the text
- "model content" "model configuration" is probably more precise

In our mind, model configuration is the assemblage of model components we use. Here, we indeed refer to model formulation, i.e., the parameterisation itself with its assumptions. It means these are the same configurations, but not the same "physical package". We agree that in terms of model, the choice of different parameterisations can be seen as a configuration, but we would like to go one step further by pointing to how a phenomenon is represented.

- At several occasions "It"/"This" is used for a couple of sentences in a row. At some point, you lose track of what the term actually refers to, which is not helpful.

  We have revisited the text with this comment in mind.
- l 28: Major aims have been to either ... and to ... Corrected
- 134: for the last glacial inception Corrected, but the sentence has also changed

- 137: What is the "initial effect of vegetation" – are you referring to the model spin-up? I wouldn't consider this a very physical motivation for vegetation feedbacks.

We are not referring to model spin-up but to the direct effect of vegetation on climate. We thus change to direct effect of vegetation on climate.

- 137: "They" – Who?

The results of these simulations. This part of the introduction has been revisited.

- 148: I guess the authors are referring to Holocene simulations? Isn't interactive vegetation common in a number of models, meanwhile, for present-day and future?

No, we are referring to model configurations used for climate projections, which is why we provided the Arias et al. 2021 reference, which is the reference of the IPCC WG1 2021 technical summary in which there is a synthesis figure about. We added the number of the Arias et al.'s figure in the reference (Figure TS2.2. In paleoclimate, most groups are convinced dynamic vegetation needs to be included, since vegetation is the first manifestation of climate change as seen in paleo pollen records...., but do not necessarily have it in the version of the models they have access to.

- 149: "still have"

Corrected

- 150: "model biases ... as those discussed by" – As the whole paper is about improving a vegetation model, it would add to the motivation of the study to name a few of the biases specifically

We added the underestimation of the green Sahara as an example

- 154: "Climate-vegetation feedbacks on climate sensitivity... in estimates of climate sensitivity"

Thank you with also added "radiative"  $\rightarrow$  "albedo and atmospheric moisture radiative feedbacks", to distinguish between the different types of feedback we are discussing.

- 1156-58: The sentence is confusing. It seems to mention the same aspect twice.

The first part is climate mean state, as the one we would consider between different climatic periods. The second one is the simulated climate mean state, i.e., it can be different between two models for the same climatic period. We have tried to clarify this point.

- 161: interconnections

Corrected

- 163: may be

Corrected

-164: "fully-coupled" and then the sentence does not make any sense to me afterwards The sentence has been revisited, and we mention model performance instead of climate system,

which is more accurate in this context.

- 167-69: The sentence is very long and convoluted

We have simplified the sentence

- 171: "the vegetation-climate feedback" – which feedback? This would be an opportunity to be more precise/expand

We specified biophysical vegetation-climate feedback

- 177: "on the model content"

**Corrected**

- 183: "We focus on estimating the atmospheric.."
- 190: remainder

Corrected

- 1105: "run using ..." → "operates on the atmospheric" Changed following the proposition

- 1119: "two parametrizations"? We decided to keep formulation here

- l182: similar orbital configuration

Corrected

- 1186/187: "It somehow provides.." Sounds a bit spongy could you be more precise? Changed to "It also provides"
- *l211: imposes a cold start for the land surface*Corrected
- 1214: "recovery" is not so much the right word here, I would suggest "adjustment"? We changed to "adjustment"
- Figure 2: There are some data gaps in panel (I), which I would not expect. Also, there is a type in "bare soil" (panel a)

Thank you for raising this point. We corrected the typo. In panel I, this was not a data gap; it was due to the lower limit on the y-axis and the fact that lines were not drawn across the x-axis. This has been corrected.

- l252: interpolated to

Corrected

- l254 and following: "centennial" would be an alternative to "100-year" We decided to keep 100-year because it corresponds to the exact size of the window we consider.
- 1265: "when accounting for uncertainties (Fig 5)" Figure 5 does not show any uncertainties Corrected to Fig. 4. The sentence has been adjusted and now starts with "The regional averages plotted in Fig. 4".
- *l266:* "as it is expected with vegetation feedback." Why is this expected? Part of the sentence was missing "larger precipitation than the IPSL standard IPSL PMIP4 simulations". We also added the reference to Braconnot et al. 1999.
- 1280: "we computed"

No change, the text is correct here.

- 1301: omit "with" Corrected

- 1329: by replacing ... one by one by those obtained ...

**Corrected**

```
- 1338: LW_sup
Corrected
- 1362: The first-order feedbacks between ... highlighted in the previous section
Corrected
- 1376: "include the change"/"benefit from the change"
Corrected
-l 384: "With the bareold scheme, ..."
Corrected
- Figure 9: "Evapnu"?; "Transpiration"
We change evapnu to bare soil evaporation
- l414: "are the differences in seasonal.."
Corrected
- 1419: "mainly originates from the relative.."
Corrected
- l431: "does not prevent"
The sentence has changed and this part has been supressed.
- 1438: "outgoing"
Corrected
- 1439: "increasing temperature ... the higher atmospheric.."
Corrected and sentence revisited
- 1444-445: This sentence does not have any meaning
For sure, the verb "are found" was missing. Corrected
- 1450: "The feedback differences between model versions"
Corrected
- l460: "To first order, the distribution ..."
Corrected
- 1498: "highlights" → "modulates" or "amplifies"?
We prefer keeping highlights here. The reason is that these feedbacks are also active when
```

dynamic vegetation is not considered, but are not necessarily seen. Of course, this is due to the fact that vegetation changes amplify the feedback because they have an effective effect on the vegetation cover, which is not the case when vegetation is prescribed.

- Figure 13: "Atmosphere", "pre-industrial" Corrected.
- 1527-528: This sentence appears superfluous to me.

Well, we keep it because it was not obvious at first glance to realise that the shape of these curves is driven by the photosynthesis parameterisation and not the distribution of vegetation.

- Figure 14: wrong panel labels for (c) and (d). Also, I was wondering about the sign convention for the NEE here – do positive values correspond to more or less update?

Corrected. The legend has been completed with the indication of the sign convention. Positive values indicate a reduction in the vegetation carbon sink.

- 1552: "photosynthesis parametrization"

Yes, parameterization is better here. Corrected

- 1558-559: I was curious if the authors could speculate about the nature of these different characteristics.

No we will not at this stage. We didn't dig deeply enough into the details of the changes in land carbon between the simulations.

- 1561: What does "direct development" imply?
- "directly on the mid-Holocene climate and not on the modern climate, as it is usually the case"
- *l575: I think the reference should be to Fig.8* Corrected

---

## Author Response (AR3)

| Dear editor,                                                                                                                                                                                        |
|-----------------------------------------------------------------------------------------------------------------------------------------------------------------------------------------------------|
| Thank you for your decision regarding this manuscript. We have changed the colours in the figures as requested to meet colour blind requirements. We have also corrected the error in the abstract. |
| Best regards                                                                                                                                                                                        |
| Pascale Braconnot, on behalf of the co-authors                                                                                                                                                      |